# Hyperreactivity to uncertainty is a key feature of subjective cognitive impairment

**Bahaaeddin Attaallah[1]\*, Pierre Petitet[2], Elista Slavkova[2], Vicky Turner[1], Youssuf Saleh[1], Sanjay G Manohar[1,2], Masud Husain[1,2]**

[1]Nuffield Department of Clinical Neurosciences, University of Oxford, Oxford, United Kingdom; [2]Department of Experimental Psychology, University of Oxford, Oxford, United Kingdom

**Abstract** With an increasingly ageing global population, more people are presenting with concerns about their cognitive function, but not all have an underlying neurodegenerative diagnosis. Subjective cognitive impairment (SCI) is a common condition describing self-reported deficits in cognition without objective evidence of cognitive impairment. Many individuals with SCI suffer from depression and anxiety, which have been hypothesised to account for their cognitive complaints. Despite this association between SCI and affective features, the cognitive and brain mechanisms underlying SCI are poorly understood. Here, we show that people with SCI are hyperreactive to uncertainty and that this might be a key mechanism accounting for their affective burden. Twenty-seven individuals with SCI performed an information sampling task, where they could actively gather information prior to decisions. Across different conditions, SCI participants sampled faster and obtained more information than matched controls to resolve uncertainty. Remarkably, despite their 'urgent' sampling behaviour, SCI participants were able to maintain their efficiency. Hyperreactivity to uncertainty indexed by this sampling behaviour correlated with the severity of affective burden including depression and anxiety. Analysis of MRI resting functional connectivity revealed that SCI participants had stronger insular-hippocampal connectivity compared to controls, which also correlated with faster sampling. These results suggest that altered uncertainty processing is a key mechanism underlying the psycho-cognitive manifestations in SCI and implicate a specific brain network target for future treatment.

**\*For correspondence:**
Bahaaeddin.Attaallah@ndcn.ox.ac.uk

**Competing interest:** The authors declare that no competing interests exist.

## Editor's evaluation

This study tests the hypothesis that subjective cognitive impairment (SCI) is linked to hyperreactivity to uncertainty. Using an information-gathering task, the authors demonstrate that individuals with SCI sample faster and more than controls under uncertainty. The reported findings provide important new clues about the psychological and neural manifestations of SCI.

## Introduction

With an ageing population, an increasing number of people are seeking medical advice for concerns about cognitive decline (*Deary et al., 2009*; *Harada et al., 2013*). While in some individuals these complaints might be related to a progressive pathological process such as Alzheimer's disease (AD), they can also be expressed by people without an underlying neurodegenerative disorder (*McWhirter et al., 2020*). When objective clinical evidence of significant cognitive impairment is not evident alongside self-reported cognitive complaints, individuals are categorised as having subjective cognitive

impairment/decline (SCI/D) (*Jessen et al., 2014*; *Jessen et al., 2020*; *Reid and Maclullich, 2006*). Although most follow a relatively benign course, a small proportion develops objective cognitive impairment and subsequently progress to dementia (*Arvanitakis et al., 2018*; *Kryscio et al., 2014*; *Mendonça et al., 2016*). Nevertheless, it remains unclear what drives subjective cognitive complaints in those people who do not have evidence of underlying neurodegeneration. Understanding the mechanisms of cognitive, behavioural, and psychiatric manifestations in SCI is thus crucial to improve clinical outcomes and enhance understanding of presentation of people who present with cognitive concerns.

A wealth of evidence suggests a particularly high prevalence of a range of mental health problems associated with SCI, in particular affective disorders such as anxiety and depression (*Hill et al., 2016*; *Hohman et al., 2011*; *Pavisic et al., 2021*; *Reid and Maclullich, 2006*). Similarly, people who primarily suffer from these psychiatric disorders often report sub-optimal cognitive performance, further emphasising the intertwined relationship between affective burden and subjective cognitive experience (*Millan et al., 2012*). Moreover, treating anxiety and depression may improve subjective cognitive complaints in individuals with SCI (*Allott et al., 2020*). Despite the association between SCI and affective burden being increasingly recognised, little is understood about the underlying cognitive mechanisms and brain networks involved.

A rich body of theoretical and empirical work suggests that affective dysregulation might be related to uncertainty processing and related behaviours (*Bishop and Gagne, 2018*; *Carleton, 2016*; *Grupe and Nitschke, 2013*; *Gu et al., 2020*). People who express higher levels of anxiety and depression often report higher levels of intolerance to uncertainty (*Boelen et al., 2016*; *Boswell et al., 2013*; *Carleton et al., 2012*; *McEvoy and Mahoney, 2011*; *Saulnier et al., 2019*). Mechanistically, intolerance to uncertainty might be reflected in several cognitive and behavioural processes underpinning goal-directed behaviour when people decide and act under uncertainty (*Grupe and Nitschke, 2013*). For example, when someone is crossing the road, they make their decision based on how confident they are that the environment is safe (i.e. they have an assessment of how uncertain the environment is for their intended action) (*Bach and Dolan, 2012*; *Gottlieb and Oudeyer, 2018*). If uncertainty is high, agents often try to reduce it by gathering information to inform their decision (e.g. checking passing cars and traffic lights and looking for a safer place to cross). People who are more sensitive to uncertainty might have an exaggerated estimation of uncertainty or preparedness when required to face it, eventually affecting their decisions and outcomes (*Grupe and Nitschke, 2013*). Similarly, uncertainty sensitivity might affect self evaluation of cognitive abilities (e.g. having lower confidence in recollection) amplifying memory complaints and subsequent emotional reaction (*Fitzgerald et al., 2017*; *Nelson, 1990*). Such a framework, which involves estimation, valuation, preparation, and learning under uncertainty allows a detailed investigation of the psychopathology of affective dysfunction (*Gottlieb and Oudeyer, 2018*; *Grupe and Nitschke, 2013*; *Sharot and Sunstein, 2020*).

Investigation of the dynamics of how people decide and act under uncertainty might hold an important key to understanding the relationship between SCI and affective dysfunction. This might be challenging to achieve using classical behavioural paradigms, for example beads task or variants of it (*Phillips et al., 1966*). These paradigms often involve randomly drawing samples from a distribution to make inferences about the distribution (e.g, deciding the predominant colour of beads in a jar). However, when people gather information to reduce uncertainty, they dynamically assess their environment and update their expectations in order to decide whether a new piece of evidence is needed and whether they can tolerate its cost (*Juni et al., 2016*; *Petitet et al., 2021*). Although the economic aspect of this behaviour has been extensively examined in previous studies (*Clark et al., 2006*; *Jones et al., 2019*; *Juni et al., 2016*), investigation into *how* information is gathered is limited. Capturing behavioural markers that might not be directly apparent using such tasks (e.g. sampling speed and efficiency) might provide important insights into underpinning mechanisms of affective disorders. This distinction has been formalised as 'active' information gathering, characterised by situations in which participants have agency over not only how much information they gather but also how information is collected in face of uncertainty (e.g. what resources to consult and when) (*Gottlieb and Oudeyer, 2018*; *Petitet et al., 2021*).

In this study, we adopted this approach using a recently developed behavioural paradigm to investigate how people with SCI decide and act (gather information) under uncertainty (*Petitet et al., 2021*). A crucial question of this study was whether uncertainty processing is associated with affective

burden. Further, to investigate the underlying brain structures and networks that might be implicated in the process, brain resting functional neuroimaging (rfMRI) data were also collected. In an *active form* of the task, participants collected informative clues, which came at a known cost, to reduce their uncertainty before committing to decisions. Crucially, they were allowed to freely gather information whenever and in whichever way they wanted. In a *passive form* of the task, this agency over uncertainty was removed. Participants were allowed only to accept or reject offers that had fixed levels of uncertainty weighed against potential reward. Decisions were made based on whether tolerating uncertainty was worth the reward on offer. This enabled us to calculate how people weigh uncertainty against reward in a passive environment where agency over uncertainty is absent. Before decisions, participants also reported their subjective uncertainty, enabling us to measure the accuracy of uncertainty estimation that might influence both active and passive behaviour.

The results from the behavioural tasks showed that individuals with SCI gather significantly more information than healthy matched controls before they commit to final decisions. They did this regardless of the cost of information and at a faster rate than controls. Despite this faster sampling, SCI participants impressively were capable of maintaining their sampling efficiency (i.e. gathering samples that were as informative as controls). This meant that they exceeded the speed-efficiency trade-off that characterises sampling behaviour of healthy controls. Crucially, in individuals with SCI, sampling speed and over-sampling (indices of heightened reactivity to uncertainty) were associated with affective burden (derived from self-report measures of anxiety and depression).

By contrast, when they had no agency over the reward and uncertainty on offer (passive choice task), SCI participants had intact metacognitive assessment and valuation of uncertainty. This suggests that controllability when dealing with uncertainty might be a crucial aspect in affective dysfunction, as differences between SCI participants and controls were apparent only when they had agency over uncertainty (i.e. during the active information gathering phase preceding decisions).

Functional neuroimaging analysis investigating whole-brain resting connectivity between regions of interest across all known brain networks revealed that individuals with SCI, in comparison to healthy controls, had increased insular hippocampal connectivity. Further, the strength of this connectivity in SCI correlated with reactivity to uncertainty indexed by sampling speed.

Taken together, the results indicate that hyperreactivity to uncertainty might be a key mechanism in SCI, and link this process to the insular cortex and hippocampus.

## Results

### Experimental design

Participants performed a recently developed behavioural task (*Petitet et al., 2021*) designed to investigate active information gathering and decision making under uncertainty (*Figure 1*). In this paradigm, participants were asked to maximise their reward by trying to localise a hidden purple circle of a fixed size as precisely as possible. They could reduce uncertainty about the location of the hidden circle by touching the screen at different locations to obtain informative clues: if a purple dot appeared where they touched, this meant that the location was situated inside the hidden circle, otherwise, the dots were coloured white. Obtaining these clues came at a cost ($\eta_s$) that participants had to pay from an initial credit reserve ($R_0$) they started each trial with. Participants could sample the search field freely without constraints to the location or the speed at which they touch the screen. At the end of each trial, participants were required to move a blue disc to where they thought the hidden circle was located. A feedback was given after this, indicating credits participants won (or lost) based on how precise their localisation was and the credits they lost to obtain information (i.e. participants had to make a trade-off between obtaining more information and the cost of this information). There were two levels of sampling cost (low and high) and two levels of initial credit reserve (low and high). Uncertainty in the task was quantified as expected error ($EE$) which is the average error that an optimal agent is expected to obtain when placing the blue disc at the best possible location. For more details see Materials and methods.

### Demographics

All participants (healthy controls and individuals with SCI) had ACE-III cognitive scores within normal performance limits (>87/100) (*Bruno and Schurmann Vignaga, 2019*; *Elamin et al., 2016*; *Hsieh et al.,*

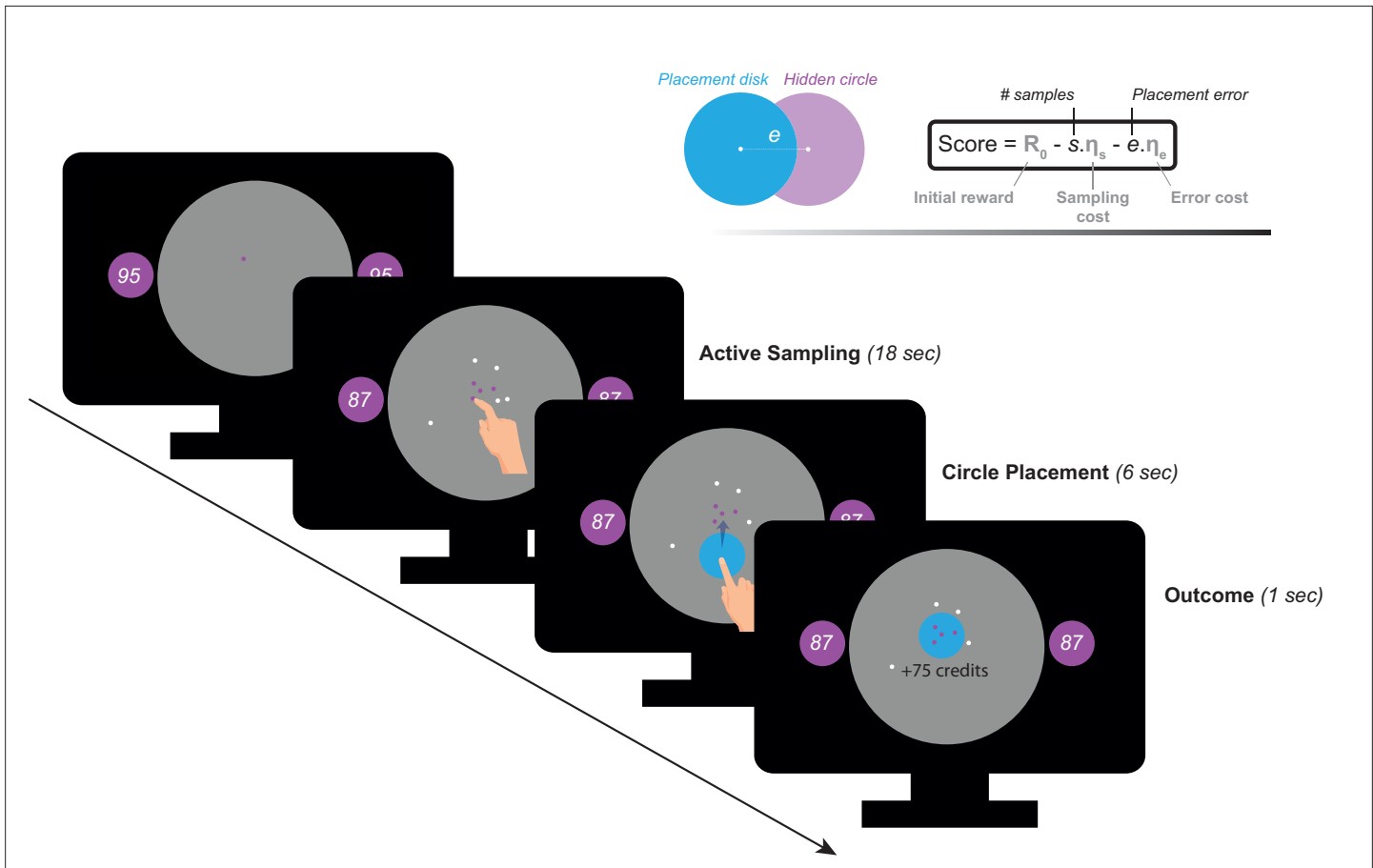

**Figure 1.** Task paradigm – Active information sampling. Participants were required to find the location of a hidden purple circle as precisely as possible. Clues about the location of the hidden circles could be obtained by touching the screen at different locations. This yielded dots that were coloured either purple or white depending on whether they were situated inside or outside the hidden circle: purple dots were inside and white dots were outside the hidden circle. Two circles of the same size as the hidden circle were always on display on either side of the screen to limit memory requirements of the task. Inside these two circles, an initial credit reserve ($R_0$) was displayed. There were two levels of $R_0$: low = 95 credits and high = 130 credits. At the beginning of each trial, a purple dot was always shown to limit initial random sampling. Participants then had 18 s on each trial, during which they could sample without restrictions to speed, location or number of samples. Each time they touched the screen to add a dot, the number of credits available decreased depending on the cost of sampling ($\eta_s$) on that trial. There were two levels of $\eta_s$: low = –1 credit/sample and high = –5 credits/sample. Once the 18 s had passed, a blue disc of the same size as the hidden circle appeared at the centre of the search field. Participants were then required to drag this disc on top of where they thought the hidden circle was located. Following this, the score they obtained on that trial was calculated and presented as feedback at the end of the trial.

*2013*). There was no significant difference between SCI participants and controls in cognitive scores (Controls: $\mu = 97.89$, $SD = 1.80$; SCI: $\mu = 95.41$, $SD = 4.21$; $z = 1.91$, $p = 0.06$). Consistent with previous reports (*Hill et al., 2016*; *Hohman et al., 2011*; *Pavisic et al., 2021*; *Reid and Maclullich, 2006*), SCI participants in the study were significantly more depressed and anxious than healthy controls (Depression: $z = 4.41$, $p < 0.001$, Anxiety: $z = 3.08$, $p < 0.01$; *Table 1* and Figure 5a.). Since depression and anxiety correlated positively with each other in both groups (Spearman's $R^2 = 0.19$, $p = 0.02$, SCI Spearman's $R^2 = 0.47$, $p < 0.001$, Figure 5b.), a principal component analysis (PCA) was performed to extract a dimension that accounts for the maximum shared variance between the two measures. This dimension could be regarded as a measure of affective burden in participants and accounted for 84% of the variance shared between depression and anxiety. Higher scores of affective burden indicate more severe depression and anxiety. There was no significant correlation between cognitive scores and this affective dimension ($p = 0.07$ & $p = 0.49$, for controls and SCI respectively, controlling for age and gender).

**Table 1.** Demographics.

| N (M/F) | Controls 27 (13/14) | | SCI 27 (13/14) | | p-value* |
|---|---|---|---|---|---|
| | **Mean** | **SD** | **Mean** | **SD** | |
| Age | 62.04 | 6.28 | 59.81 | 7.70 | 0.34 |
| ACE-III | 97.89 | 1.80 | 95.41 | 4.21 | 0.06 |
| BDI-II | 4.59 | 4.36 | 15.44 | 11.24 | <0.001 |
| HADS Dep. | 1.48 | 1.81 | 5.26 | 4.61 | <0.001 |
| HADS Anx. | 4.30 | 3.16 | 7.04 | 3.32 | <0.01 |

ACE-III: Addenbrooke's Cognitive Examination. BDI-II: Beck Depression Inventory. HADS: Hospital Anxiety Depression Scale.
* Student-test or Wilcoxon rank-sum test for parametric and non-parametric data, respectively.

## Extensive sampling in SCI

As prescribed by rational behaviour, participants in both groups (SCI and healthy controls) adjusted the extent of their search ($s$) to the sampling cost ($\eta_s$), acquiring fewer samples when this cost increased (Effect of $\eta_s$ on $s$, *Figure 2a.*, *Supplementary file 1*). While there was no significant main effect of initial credit ($R_0$), its interaction with sampling cost was significant ($\beta = 0.02$, $95\%CI = (-0.04, -0.004)$, $t_{3232} = 2.41$, $p = 0.016$, *Supplementary file 1*), which means that the aversive effect of sampling cost on the number of samples obtained was blunted when participants started their search with a larger credit reserve (*Figure 2a.*).

The influence of these economic features ($\eta_s$ and $R_0$) on the number of samples acquired was not significantly different between SCI participants and controls ($SCI \times \eta_s : p = 0.13; SCI \times R_0 = 0.27$). Nevertheless, overall, individuals with SCI sampled significantly more than controls (Main effect of SCI on $s: \beta = +0.19$, $95\%CI = (0.058, 0.32)$, $t_{3232} = 2.81$, $p < 0.01$, *Figure 2a.*, *Supplementary file 1*). Gathering more samples led SCI participants to finish their active search at lower levels of uncertainty ($EE$) than controls on average (Main effect of SCI on the $EE$ reached at the end of the sampling phase: $\beta = 0.26$, $95\%CI = (-0.453, -0.06)$, $t_{3232} = 2.66$, $p < 0.01$, *Supplementary file 1*), which translated into smaller localisation errors (Main effect of SCI on localisation error: $\beta = -0.18$, $95\%CI = (-0.35, -0.004)$, $t_{3232} = -2.01$, $p = 0.045$).

Next, we asked whether SCI participants' more extended information gathering led to better performance. To answer this question, we calculated, on each trial, the optimal number of samples, $s^\star$, that maximises the expected value of the trial ($EV$). Both acquiring extra samples beyond this point (i.e. over-sampling) and not sampling enough to reach it (i.e. under-sampling) result in a smaller expected value (*Figure 2—figure supplement 2*). Thus, this analysis provided some insight into the usefulness of the extensive sampling behaviour SCI participants exhibited compared to controls.

Both healthy controls and individuals with SCI over-sampled relative to the optimal stopping point when the sampling cost was high ($p < 0.001$ for both groups, see *Supplementary file 3* for statistical details). Consistent with above, over-sampling in these conditions was significantly more pronounced in SCI participants compared to controls (Group difference in ($s - s^\star$) at high $\eta_s$; Low $R_0$: $t(52) = 2.066$, $p = 0.04$, High $R_0$: $t(52) = 3.32$, $p < 0.01$). Thus, SCI participants' tendency to gather more information than controls in these conditions led them to acquire samples with a price outweighing their instrumental benefit.

By contrast, when the sampling cost was low, controls under-sampled relative to the optimal solution ($p < 0.001$ for the two conditions with low $\eta_s$, see *Supplementary file 3*). Thus, because they acquired more samples in these conditions too, SCI participants better approached optimal sampling behaviour (Group difference in ($s - s^\star$) at low $\eta_s$; Low $R_0$: $t(52) = 3.29$, $p < 0.01$, High $R_0$: $t(52) = 3.7$, $p < 0.001$; *Figure 2c.*).

To summarise, individuals with SCI sampled more than controls across all experimental conditions, regardless of economic constraints. This was sub-optimal when sampling was expensive (i.e. they overpaid for information) but led to more optimal behaviour when sampling was cheap.

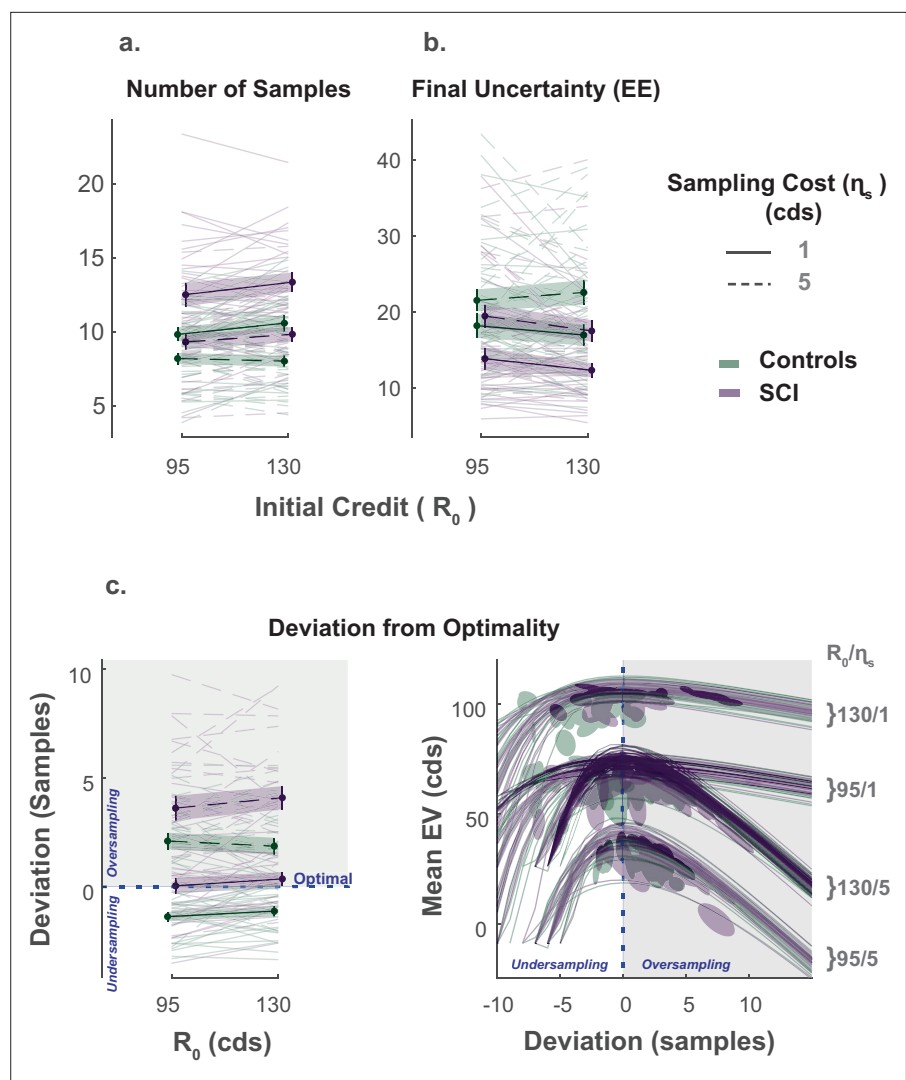

**Figure 2.** Extensive sampling in SCI. (**a**) Across different conditions, individuals with SCI sampled more than healthy controls. (**b**) Consequently, SCI participants reached final uncertainty levels (*EE*) lower than controls prior to committing to decisions. (**c**) Healthy controls and individuals with SCI over-sampled when sampling cost was high. Over-sampling was more significant in SCI than healthy controls. When sampling cost was low, healthy controls under-sampled while SCI participants were optimal. The panel on the bottom right depicts the changes in expected value (*EV*) as a function of the number of samples deviating from optimal. The optimal number of samples is when *EV* is maximum. Error bars show ± SEM. See *Supplementary file 1*, *Supplementary file 2* and *Supplementary file 3* for full statistical details.

The online version of this article includes the following figure supplement(s) for figure 2:

**Figure supplement 1.** Expected error as a function of sampling.

**Figure supplement 2.** Optimal sampling.

## Intact passive decision making in individuals with SCI

A passive version of the paradigm was used to investigate what drove SCI participants' extensive sampling behaviour. More specifically, we tested two hypotheses. First, SCI individuals might have inflated subjective estimates of uncertainty. If this were the case, they might need to reduce uncertainty to a greater extent in order to reach comparable subjective uncertainty levels. Second, SCI participants might have intact estimation of uncertainty but nonetheless assign greater weight to it when balancing it against reward. To test these hypotheses, SCI participants and healthy controls performed a modified version of the paradigm in which they were required to first, report their

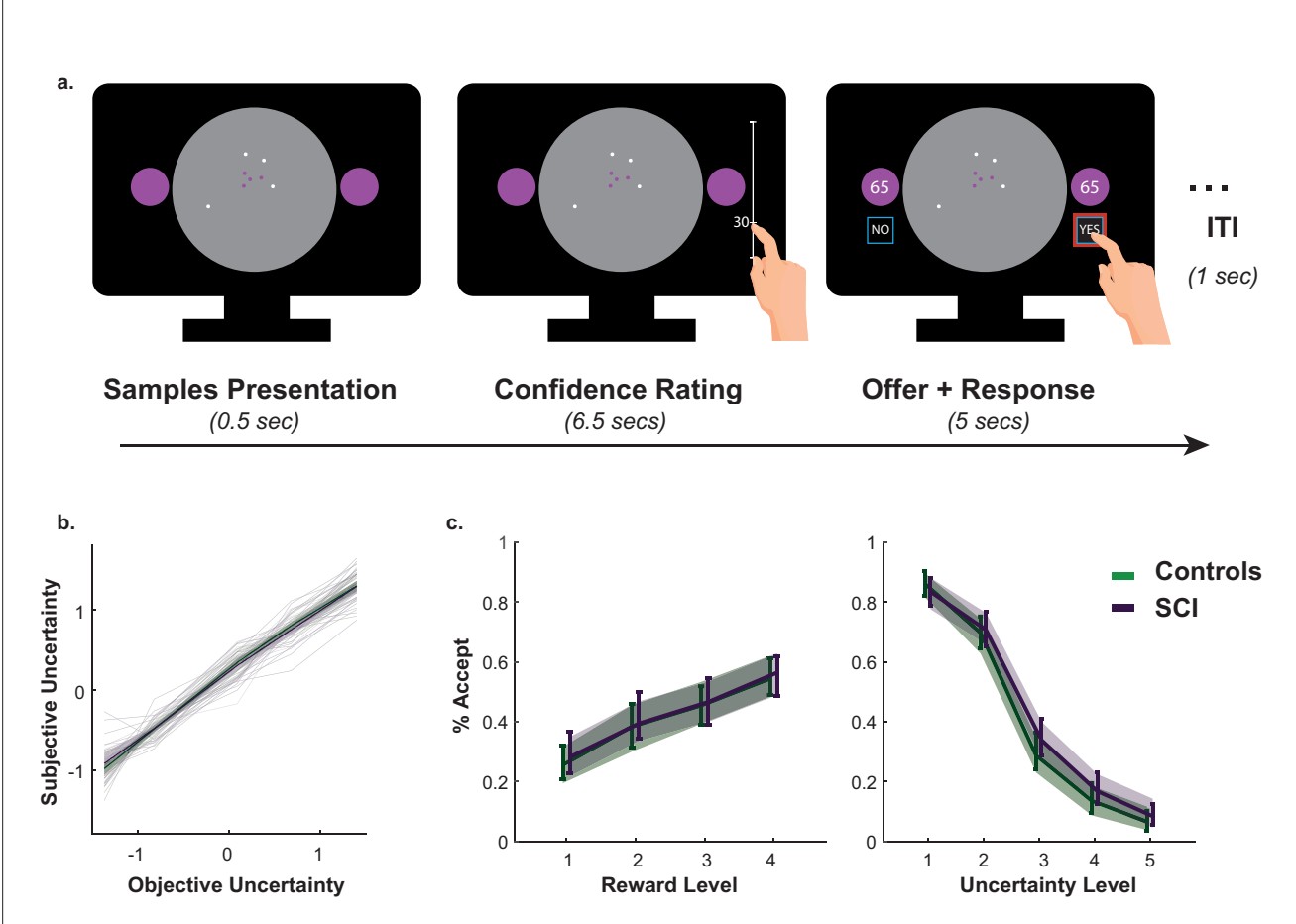

**Figure 3.** Intact metacognitive judgement and passive decision making in SCI. (**a**) Subjective estimates of uncertainty (z-scored sign-flipped confidence ratings) mapped well onto experimentally defined uncertainty across study participants. There was no significant difference between SCI participants and controls in estimating uncertainty. (**b**) There was no significant difference in offer acceptance between individuals with SCI and controls, indicating similar weights assigned to uncertainty and reward when making decisions. Error bars show 95% CI. See *Supplementary files 4 and 5* for statistical details.

estimations of experimentally defined levels of uncertainty and second, to accept/reject offers based on whether reward on offer is worth the risk imposed by uncertainty (*Figure 3a.*).

Generalised mixed effects model was used to investigate the differences in subjective uncertainty between SCI participants and controls. This showed no significant difference in this measure between the two groups (Interaction Group × EE: $\beta = +0.001$, $95\%CI = (-0.12, 0.11)$, $t_{5396} = -0.03$, $p = 0.98$, *Figure 3b.*, *Supplementary file 4*), suggesting that the tendency to sample more in the active experiment was unlikely caused by biased subjective estimates of uncertainty (i.e. there is no difference in the perceived informational utility of the samples). Similarly, there was no significant difference between the two groups in offer acceptance or in the effects of uncertainty and reward on offer acceptance (Main effect of SCI on offer acceptance: $\beta = +0.029$, $95\%CI = (-0.75, 0.80)$, $t_{5392} = +0.07$, $p = 0.94$; SCI interaction with reward and uncertainty: $p = 0.70$ & $p = 0.60$, respectively, *Figure 3c.*, *Supplementary file 5*). This is consistent with the finding that SCI participants' extensive sampling behaviour in the active paradigm was mostly independent from economic constraints (no significant interaction SCI$\times R_0$ or SCI$\times \eta_s$; *Figure 2*).

Taken together, these results indicate that extensive sampling in SCI is not related to the way individuals estimate or value uncertainty. Instead, it is likely to capture an intrinsic drive to gather information specifically when SCI have agency over the level of uncertainty (i.e. during active sampling) (see *Appendix 1* for a computational model capturing this effect).

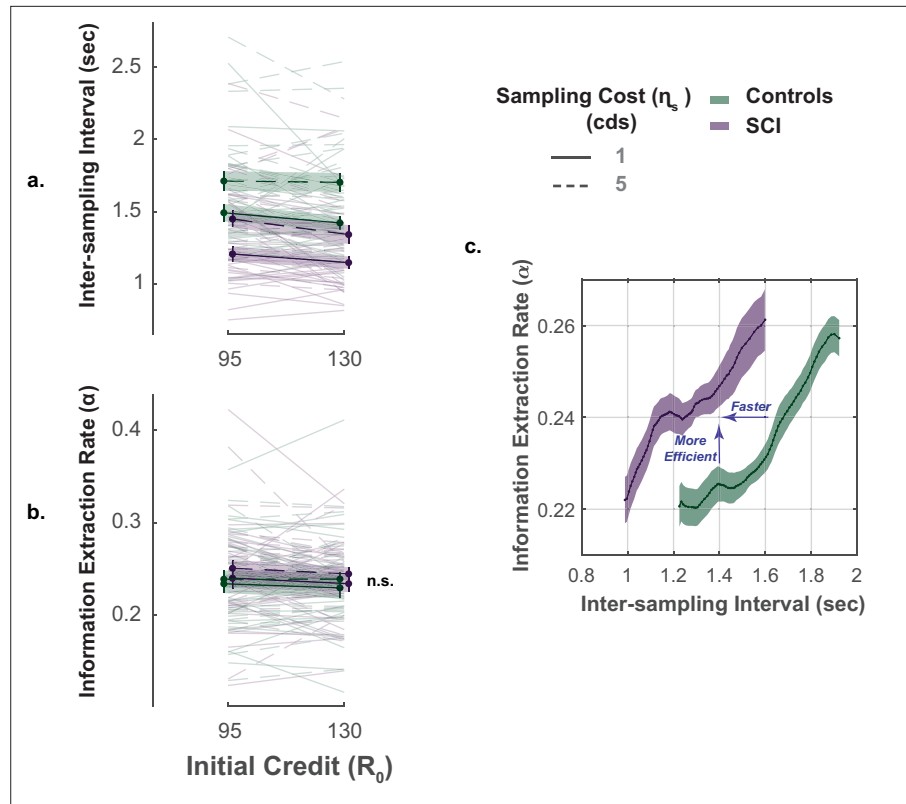

**Figure 4.** Faster but efficient sampling in SCI. (**a**) Across different conditions of the task, SCI participants sampled faster than the healthy controls. (**b**) Sampling efficiency was not different between individuals with SCI and control. (**c**) Faster sampling was associated with lower efficiency giving rise to a speed-efficiency trade-off. SCI participants exceeded the speed efficiency trade-off that characterised controls' sampling behaviour as they extracted more information than the control per unit time (sec). Conditional plot was generated by sliding 25% quantile bins of *ISI* and computing average $\alpha$ for each bin. Error bars show ± SEM. See ***Supplementary file 6*** for full and statistical details.

The online version of this article includes the following figure supplement(s) for figure 4:

**Figure supplement 1.** Sampling as a function of time.

## Faster and more efficient sampling in SCI

The key advantage of our paradigm is the possibility to investigate not only how much information people gather but also how quickly and efficiently they do so (***Petitet et al., 2021***). To capture these extra dimensions of sampling behaviour, we used two behavioural measures: (1) inter-sampling interval, *ISI*, which is the average time interval between consecutive screen touches (shorter *ISI* indicates faster sampling); (2) information extraction rate, $\alpha$, which provides an estimate of the rate at which the *EE* decays over successive samples (higher $\alpha$ values indicate higher sampling efficiency).

In young healthy adults, we previously reported a speed-efficiency trade-off whereby slower sampling was associated with greater information extraction rate (i.e. greater reduction of uncertainty at each step of the search) (***Petitet et al., 2021***). This finding was replicated in the present study (Effect of *ISI* on $\alpha$; Controls: $\beta = +0.054$, $95\%CI = (0.032, 0.075)$, $t_{1618} = 4.97$, $p < 0.0001$; SCI: $\beta = +0.052$, $95\%CI = (0.028, 0.076)$, $t_{1618} = 4.97$, $p < 0.0001$, ***Figure 4c.***, ***Supplementary file 8***). Investigating group effect using LMM showed that, overall, SCI participants sampled significantly faster than healthy controls (Main effect of group on *ISI* : $\beta = -0.29$, $95\%CI = (-0.43, -0.16)$, $t_{3232} = -4.24$, $p < 0.0001$, ***Figure 4a.***, ***Figure 4—figure supplement 1***, ***Supplementary file 6***). Remarkably, despite this faster sampling, SCI participants reduced uncertainty as efficiently as controls (Main effect of group on $\alpha$ : $\beta = +0.007$, $95\%CI = (-0.015, 0.03)$, $t_{3232} = 0.61$, $p = 0.54$, ***Figure 4b.***, ***Supplementary file 6***). In other words, individuals with SCI exceeded the speed-efficiency trade-off that characterised healthy controls' sampling (***Figure 4c.***).

Overall, the performance of individuals with SCI indicates *hyperreactivity to uncertainty*, manifested as more extended, faster though equally efficient information sampling compared to controls.

### Affective burden is associated with more extensive and faster active sampling

Next, we investigated whether markers of hyperreactivity to uncertainty in SCI (faster and extensive sampling) were associated with affective burden. Non-parametric Spearman's partial correlations

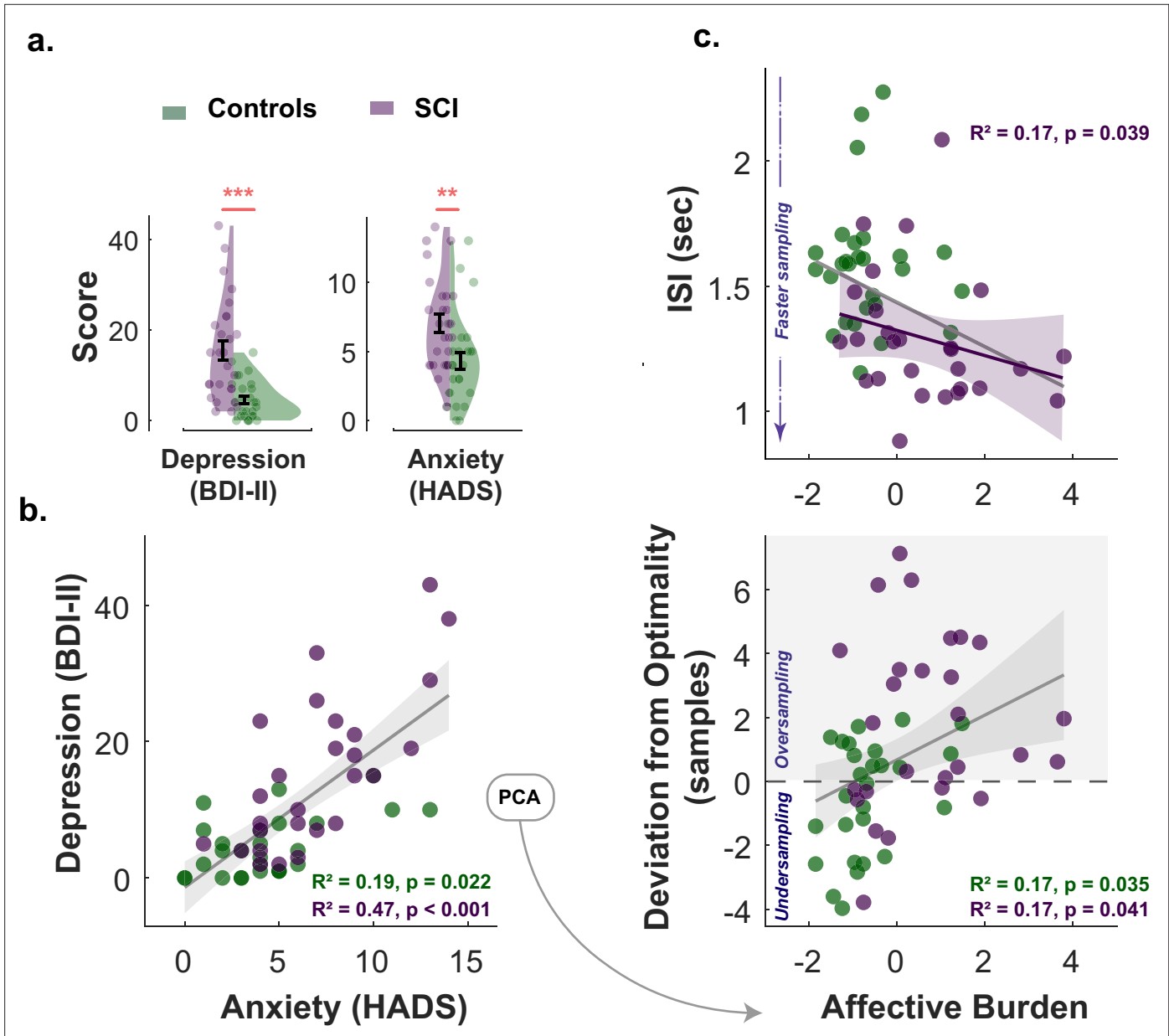

**Figure 5.** Affective burden correlates with faster and extensive sampling. (**a**) Individuals with SCI were significantly more depressed and anxious than healthy matched control. (**b**) Anxiety scores and depression scores significantly correlated with each other across study participants of both groups. An affective burden score corresponding to severity of depression and anxiety was extracted using principal component analysis (PCA). This dimension accounted for 84% of the variance between anxiety and depression. (**c**) Affective burden was associated with increased reactivity to uncertainty indexed by speed and extent of sampling. More severe affective burden (i.e. higher severity of anxiety and depression) were associated with faster (shorter *ISI*) and more extensive sampling (i.e. over-sampling). BDI II: Beck's Depression Inventory. HADS: Hospital Anxiety and Depression Scale (only anxiety score was included). ISI: Inter-sampling Interval in seconds. $**: p < 0.01$, $***: p < 0.001$. Error bars show ± SEM. Grey and purple lines show regression across all participants and within SCI group, respectively. Shaded area in correlation plots show 95% CI.

controlled for age and cognitive score showed that affective burden across SCI participants was significantly associated with sampling speed as well as with deviation from optimal sampling, such that both faster and extensive sampling were associated with higher affective burden (Correlation between affective burden and $ISI : R^2 = 0.17$, $p = 0.039$; correlation between affective burden and $s - s^\star : R^2 = 0.17$, $p = 0.041$, *Figure 5c.*). A similar pattern was also observed in the control group for over-sampling behaviour ($R^2 = 0.17$, $p = 0.035$) but not for $ISI$ ($p = 0.78$).

To assess the individual contribution of anxiety and depression to these findings in SCI, the same analysis was performed separately for each affective component. This showed that BDI-II score (depression) correlated with faster sampling ($R^2 = 0.17$, $p = 0.039$) but not over-sampling ($p = 0.15$) in SCI. The HADS score (anxiety) on the other hand correlated with over-sampling ($R^2 = 0.17$, $p = 0.038$) but not speed ($p = 0.16$). However, these effects became insignificant ($p = 0.12$ for both) when the two affective components partialled out each other (partial correlations), suggesting that while there might be distinctive effects of depression and anxiety, these effects are better explained by the general affective burden imposed by these two dimensions given their association.

Finally, the correlation between affective burden and passive task performance was investigated using LMMs. The results showed that affective burden had no significant effect (main or interaction) on subjective uncertainty or acceptance probability (Main effect of affective burden on subjective uncertainty: $\beta = -0.003$, $t_{5394} = -0.27$, $p = 0.79$; Main effect of affective burden on offer acceptance: $\beta = -0.151$, $t_{5390} = -1.96$, $p = 0.05$; See *Supplementary files 4 and 5* for full statistical details including interactions).

These results thus indicate that hyperreactivity to uncertainty (indexed by faster and extensive sampling) is associated with affective dysregulation in SCI, mainly when agency is involved.

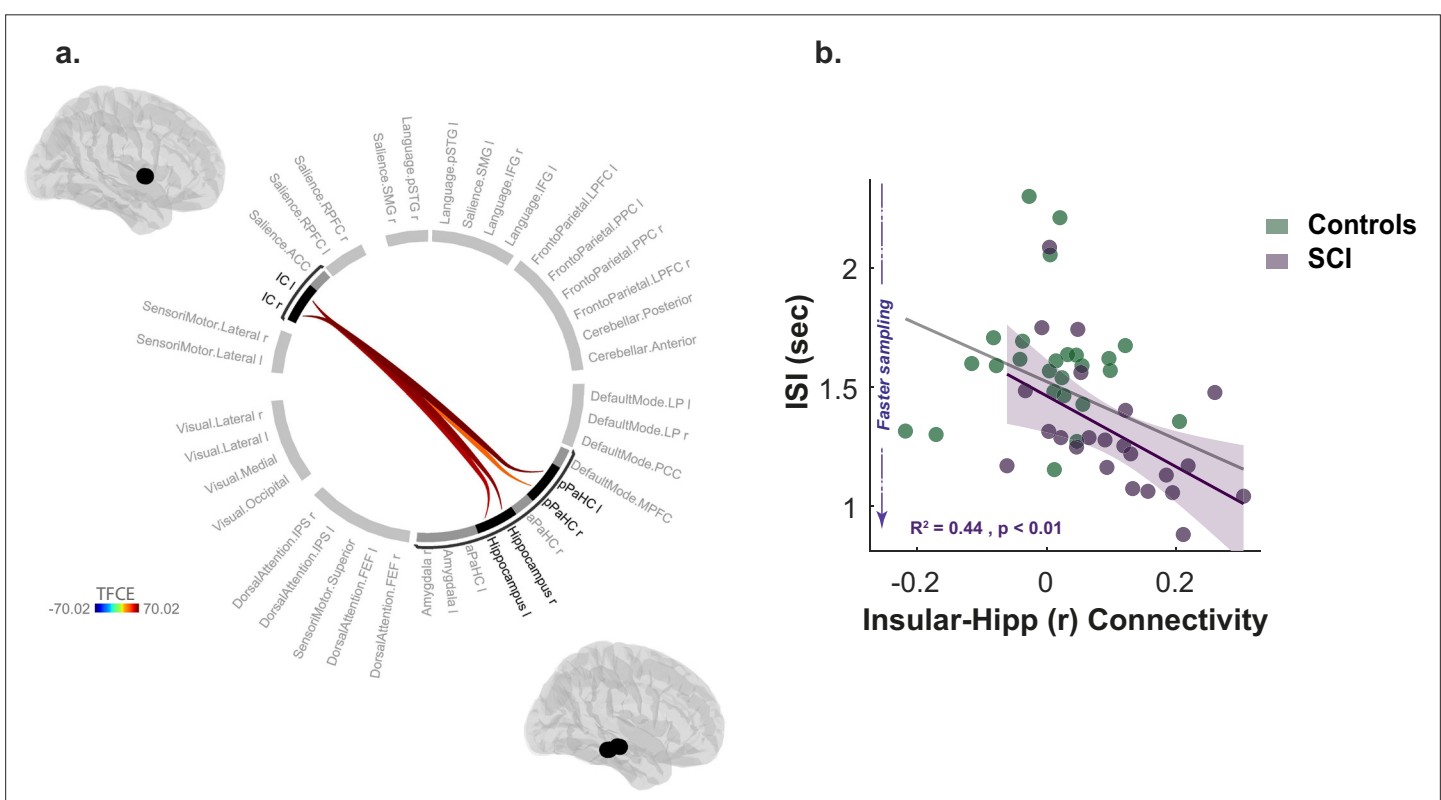

**Figure 6.** Increased insular hippocampal connectivity in individuals with SCI compared to healthy controls. (**a**) Whole-brain ROI-to-ROI functional connectivity analysis with 40 ROIs from different brain networks and regions. SCI participants have increased functional connectivity between insular cortex (IC) and hippocampal/para-hippocampal (PaHC) regions. TFCE: Threshold Free Cluster Enhancement. See *Supplementary file 9* for further statistical details. (**b**) Increased insular-hippocampal connectivity correlated with faster sampling (shorter *ISI*) in SCI, suggesting that hyperreactivity to uncertainty is related to this specific network. Grey and purple shaded line show regression across all participants and within SCI group, respectively. Shaded area show 95% CI.

## Increased insular-hippocampal connectivity in SCI is associated with faster sampling

Resting-state functional MRI data were collected in 23 SCI participants and 25 controls. We first investigated differences in functional connectivity between SCI participants and controls using a whole-brain network functional connectivity analysis. This entailed investigating the connections between 40 atlas-defined regions of interest (ROIs) that represent the key nodes in brain networks including salience, default mode, sensorimotor, visual, dorsal attention, frontoparietal, language, and cerebellar networks as well as limbic brain regions including the hippocampus, para-hippocampus and amygdala (see Materials and methods).

Compared to healthy controls, individuals with SCI showed significantly greater functional connectivity between the insular cortex and hippocampal/para-hippocampal regions (Insular-hippocampal: $TFCE = 64.82$, $p_{FWE} < 0.01$, Insular-para-hippocampal: $TFCE = 70.02$, $p_{FWE} < 0.01$; *Figure 6a*, *Supplementary file 9*). There was no other significant difference in resting functional connectivity between any of the other ROIs included in the analysis. Thus, functional connectivity disruptions in SCI participants appeared to be limited and specific to the insular-hippocampal network highlighted in *Figure 6a*.

Finally, we asked whether such functional signatures of SCI bore any relationship to the behavioural markers of hyperreactivity to uncertainty identified above (*ISI* & $s - s^\star$). This analysis used non-parametric partial correlations controlling for age, gender, and cognitive score. The strength of connectivity from ROI-ROI analysis between bilateral insular cortex and hippocampus (right and left) was extracted for this purpose. Multiple correlation testing was corrected with Bonferroni's method. Across SCI participants, stronger bilateral insular connectivity with the right hippocampus significantly correlated with faster sampling rate ($R^2 = 0.44$, $p_{corr} < 0.01$, *Figure 6b*). This suggests a role for this network over-activity in the urgent information sampling behaviour exhibited by SCI participants. There was no significant association with deviation from optimal sampling ($p = 0.19$), suggesting that sampling speed might be a more sensitive marker of uncertainty reactivity, despite that these two measures were significantly correlated in the group ($R^2 = 0.28$, $p < 0.01$). Similarly, insular-hippocampal connectivity was not significantly correlated with affective burden across SCI participants ($p = 0.37$).

## Discussion

A rich body of literature indicates a strong association between SCI and affective dysregulation such as anxiety and depression (*Hill et al., 2016*; *Hohman et al., 2011*; *Pavisic et al., 2021*; *Reid and Maclullich, 2006*). However, the mechanisms underlying such a burden are not fully established. In this study, we hypothesised that affective dysfunction in SCI might be related to deficits in processing uncertainty. Using a purpose-designed behavioural paradigm, we investigated how people decide and act under uncertainty in active and passive contexts. In the active form, participants could gather information at a cost to reduce uncertainty before committing to decisions. In the passive form, they made decisions responding to offers that had fixed levels of uncertainty and potential reward. The results showed that when participants had agency (i.e. in the active form), individuals with SCI exhibited behaviour indicative of increased reactivity to uncertainty, manifested as more rapid and extensive sampling compared to age- and gender-matched healthy controls. These behavioural markers of heightened reactivity to uncertainty correlated with the severity of affective burden in SCI participants. Furthermore, resting functional neuroimaging analysis showed that individuals with SCI had increased insular-hippocampal connectivity, which in turn correlated with reactivity to uncertainty indexed by sampling speed. By contrast, estimation and weighing of uncertainty in passive decisions were both intact in SCI participants, suggesting that their hyperreactivity to uncertainty was specifically expressed in an active situation in which uncertainty is controllable. Overall, the results point to a specific deficit in processing uncertainty in individuals with SCI that might be underlying their affective dysregulation and is related to increased insular-hippocampal connectivity in the condition.

These results resonate with previous research indicating that people with depression and anxiety might have altered uncertainty processing (*Bishop and Gagne, 2018*; *Boswell et al., 2013*; *Carleton et al., 2012*; *Grupe and Nitschke, 2013*; *Gu et al., 2020*; *Hartley and Phelps, 2012*; *Pulcu and Browning, 2019*; *Saulnier et al., 2019*). A recent investigation, for example, showed that a common factor accounting for shared variance between both syndromes was associated with disrupted learning in probabilistic environments reflecting impaired uncertainty processing (*Gagne et al., 2020*). Other

reports pointed to a possible alteration of uncertainty estimation and reward-related valuation in these syndromes affecting decision making under uncertainty (*Pizzagalli et al., 2005*; *Pizzagalli et al., 2008*; *Pulcu and Browning, 2019*). However, evidence on how affective dysfunction relates to more active forms of behaviour such as information gathering prior to committing decisions is limited. One early investigation in social psychology showed that individuals suffering from depression tend to acquire more high-utility information than non-depressed individuals in a simulated interview environment where participants played the role of the interviewer and had to select interview questions from a standardised list of questions that differed in their diagnostic utility (*Hildebrand-Saints and Weary, 1989*). While this investigation along with other similar reports from social psychology literature advance the notion that disrupted information gathering might be a key feature in affective disorders, a mechanistic account supporting this claim is still not established (*Aderka et al., 2013*; *Camp, 1986*; *Joiner et al., 2009*; *Locander and Hermann, 1979*). For example, previous studies using classical behavioural paradigms of information seeking such as the beads task failed to report consistent effects of anxiety and depression on performance (*Jacoby et al., 2014*). Such inconsistencies might be due to the fact that the behavioural paradigms used in these prior studies often neglect an important aspect of information gathering – controllability – which has been hypothesised to be a crucial feature of both anxiety and depression (*Abramson et al., 1978*; *Barlow, 1991*). In the present study, this important component was accounted for by using a novel behavioural paradigm allowing participants to gather information with minimal limitations to the speed and efficiency of sampling. This in turn revealed an insightful aspect of information gathering behaviour in individuals with SCI who not only sampled more than controls, but also showed that they do this more rapidly without losing efficiency.

One might argue that rather than suggesting that SCI individuals are hyperreactive to uncertainty in the active task, our findings might alternatively be explained by a lower reactivity to reward – as participants lost more credits to obtain the extra information. However, some observations in the study suggest that this might not be the case. First, the influence of economic constraints ($R_0$, $\eta_s$) on sampling behaviour and speed did not significantly differ between individuals with SCI and controls. If SCI participants were less reactive to reward, then one would expect economic constraints to affect sampling behaviour in SCI participants to a lesser degree than age- and gender-matched controls. Instead, the group effect (i.e. acquiring more samples in individuals with SCI) was equally manifested in all experimental conditions, regardless of the current cost-benefit structure. Second, acquiring more samples indeed allowed SCI participants to achieve lower uncertainty levels prior to decisions, suggesting that these additional samples carried instrumental utility and were not merely reflective of wasteful sampling behaviour driven by insensitivity to reward.

Another account of SCI individuals' extensive sampling behaviour might involve subjective costs that are not directly specified in economic terms in the task. Namely, SCI individuals and controls might differ in the subjective cost they assign to sampling speed and efficiency. As demonstrated previously, these additional agent-related factors contribute to the utility of the samples being acquired and thus influence the extensiveness of sampling (*Petitet et al., 2021*). Given that SCI participants sampled faster but as efficiently as controls, these subjective costs appear to be lower in this population (see *Appendix 1* for a formal demonstration using a computational model). As a result, samples carry higher utility overall, which promotes more extended data gathering. Potentially, there are several other subjective costs that people might be considering when gathering information. For example, SCI participants might have exaggerated cost linked to bad performance or inaccurate placements of the circle. We aimed to limit this specific cost by not displaying the hidden circle as feedback and showing the credits won or lost instead (*Figure 1*). In addition, since the task depends on touching the screen to obtain clues, the cost of these motor actions might have influenced sampling behaviour (*Carland et al., 2019*; *Cisek and Kalaska, 2010*; *Morel et al., 2017*; *Pierrieau et al., 2021*; *Rangel and Hare, 2010*; *Scott, 2012*). It is possible, for example, that SCI participants might be regarding reaching and touching the screen as less effortful than controls or assigning lower costs for motor precision. While we tried to limit the influence of the motor aspects on performance by strictly controlling the settings (e.g. fixed distance from the screen, using only index finger of dominant hand), the contribution of motor elements to the observed behaviour cannot be entirely determined or excluded and might pose a limitation. Such inter-individual differences could be in part captured by the intercept term in the computational model used to fit active

information sampling data ($w_0$), which was not significantly different between the two groups (see *Appendix 1*).

In a similar vein, while the results from the passive task suggest that SCI individuals have intact estimation of uncertainty, this might not necessarily apply to a dynamic situation such as the active task. The distinct contexts of the two tasks (passive viewing vs. active gathering) may impose different computations involved in uncertainty estimation. For example, SCI individuals and controls might specifically differ in how they judge self-generated configurations of uncertainty (active task) vs. externally predefined ones (passive task) (*Acosta, 1982*; *Gaschler et al., 2014*; *Kemper et al., 2012*; *Kemper and Gaschler, 2017*). People might also differ in the reference points (temporal or spatial) they compare updated environments with as more information is added (*Koop and Johnson, 2012*; *O'Donoghue and Sprenger, 2018*; *Sher and McKenzie, 2006*; *Tversky and Kahneman, 1986*). It is also possible that these computations might be cognitively effortful, influencing how frequently or/and efficiently people re-estimate and update uncertainty during their search (*Bhui et al., 2021*; *Lieder and Griffiths, 2019*). All these factors might have contributed with different degrees to the pattern of the results observed in the study and might be a limitation to active information sampling investigations in general.

It should also be noted that the measure of subjective uncertainty based on self-reports in the passive may map imperfectly onto actual internal uncertainty (*Navajas et al., 2017*; *Peters, 2022*). Self-reports of confidence might be contaminated by other factors such as perceptual uncertainty (e.g. meaning of perceived stimuli on the task), memory (e.g. remembering task instructions or previous estimations) and decisions (e.g. confidence in estimation). The contribution of these meta-cognitive dimensions could be explored in future studies, especially given that SCI – by definition – is characterised by inconsistency between self-report estimates and objective measures (*Jessen et al., 2020*). Similarly, SCI might involve different cognitive domains (e.g. memory, attention, language, etc.) with different levels of severity (*Diaz-Galvan et al., 2021*; *Jessen et al., 2020*; *Miebach et al., 2019*; *Si et al., 2020*; *Smart et al., 2014*). How such differences relate to uncertainty processing and affective burden will require a fine-grained approach aimed specifically at characterising SCI cognitive complaints in-depth to disentangle the dimensions involved.

Various brain regions have previously been implicated in anxiety and depression and their mechanistic characterisation as deficits in uncertainty processing including amygdala (*Grupe and Nitschke, 2013*; *Morriss et al., 2019*), hippocampus (*Gray and McNaughton, 2003*; *Harrison et al., 2006*; *Rigoli et al., 2019*; *Strange et al., 2005*; *Tobia et al., 2012*), and insular cortex (*Grupe and Nitschke, 2013*; *Morriss et al., 2019*; *Tanovic et al., 2018*). Consistent with these reports, we found that individuals with SCI displayed heightened connectivity between these regions (insular-limbic). Conceptually, the insula stands out in the context of SCI not only because of its consistent implications in various forms of uncertainty processing and affective syndromes in health and disease (*Morriss et al., 2019*; *Namkung et al., 2017*; *Paulus and Stein, 2006*; *Singer et al., 2009*; *Tanovic et al., 2018*), but also because of its potential role in subjective awareness and interoception (*Craig, 2009*). The insular cortex receives input from different brain regions carrying interoceptive information about various bodily sensations, such as temperature, heartbeat, bowel distension and more (*Namkung et al., 2017*; *Uddin et al., 2017*). Subjective experience of these stimuli has been shown to correlate with insular activity on functional neuroimaging using MRI or positron emission tomography (*Craig et al., 2000*; *Critchley et al., 2004*). More recent accounts of insular function extend this contribution of subjective awareness to involve emotional states and higher subjective awareness (*Chang et al., 2013*; *Craig, 2009*; *Namkung et al., 2017*). It is thus not surprising to find insular involvement in a condition that is primarily defined by altered subjective experience. A few studies have demonstrated altered insular task-related activity in SCI linked to impaired memory performance and future-guided decision making, however, without pointing to how these findings are related to affective burden or subjectivity (*Cai et al., 2020*; *Hu et al., 2017*). By contrast, damage to the insula might impair subjectivity and self-awareness as seen in patients with anosognosia for hemiplegia and other forms of insular injury (*Fotopoulou et al., 2010*; *Karnath et al., 2005*; *Spinazzola et al., 2008*), as well as resulting in dysfunctional emotional awareness (e.g. as seen in fronto-temporal dementia), under-reactivity, lack of self-monitoring and passivity (*Kleiner et al., 2007*; *Manes et al., 1999*; *Sturm et al., 2006*). Such findings are opposite to what is observed in individuals with SCI who often report heightened levels of subjective affective dysfunction and were found in this study to be behaviourally more reactive.

This insula-centred formalisation of affective dysfunction, uncertainty processing, and subjectivity might be further supported by taking into consideration hippocampal contribution. One prominent view of hippocampal involvement in goal-directed behaviour suggests that it constitutes a crucial part of a behavioural inhibition system (BIS) that is concerned with processing aversive cues such as uncertainty (*Gray and McNaughton, 2003*). According to this view, the hippocampus acts as a comparator between one's expectations and the environment, resulting in behavioural aversion to threatening and negative stimuli. It has therefore been suggested that hyperactivity of the BIS might be an important neurobiological basis of anxiety and affective dysregulation (*Gray and McNaughton, 2003*). When anticipating decisions and actions under uncertainty, the hippocampus might be encoding possible future states and their associated risks (*Addis et al., 2011*; *Martin et al., 2011*; *Schacter et al., 2008*; *Schacter et al., 2017*; *Weiler et al., 2010*). However, when such situations are avoidable (as in the active task), aversion might be expressed as a propensity to quickly collect information to avoid facing uncertainty in the future. Hippocampal signals encoding future trajectories and prior experiences from these contexts might be shared with the insula, which in turn process emotional responses resulting in a feeling of anxiety and depression (*Chang et al., 2013*; *Craig, 2009*; *Paulus and Stein, 2006*). Additionally, because of the hippocampus' well-established mnemonic function, insular-hippocampal processing of uncertainty might also affect one's awareness of memory performance, giving rise to subjective cognitive complaints characterising SCI. Testing such a hypothesis will require future research with a detailed examination of the nature of subjective complaints, their severity, and association with uncertainty and expression of concerns.

These neuroimaging findings are consistent with previous fMRI studies (task-related and resting) in SCI showing altered connectivity between temporal brain regions (e.g. hippocampus and parahippocampus) and several other areas and networks including default mode and salience networks (*Cai et al., 2020*; *Dillen et al., 2017*; *Hafkemeijer et al., 2013*; *Hu et al., 2017*; *Rodda et al., 2009*; *Verfaillie et al., 2018*; *Viviano and Damoiseaux, 2020*). These investigations have reported both increased and decreased activation in these networks without converging spatial specificity or clear relationship with behaviour. While the hippocampus seems to be a key region commonly involved in SCI by many fMRI studies, a consistent pattern of its activity in the condition is still lacking (*Dillen et al., 2017*; *Hafkemeijer et al., 2013*; *Rodda et al., 2009*; *Verfaillie et al., 2018*). This might be due to issues related to sample size, methods used and importantly, phenotyping of SCI group included in such studies.

A recent account of these fMRI studies has tried to reconcile these apparent inconsistencies by promoting the idea that SCI might be related to general inefficiency in signal processing across whole-brain networks (*Viviano and Damoiseaux, 2020*). Our findings, however, point to a specific hippocampus-related network that might be involved in uncertainty processing and affective regulation. This could have future theoretical implications in SCI as converging reports support the role of regions of this network (limbic region and insula) in uncertainty processing and affective functioning (*Gray and McNaughton, 2003*; *Grupe and Nitschke, 2013*; *Harrison et al., 2006*; *Morriss et al., 2019*; *Rigoli et al., 2019*; *Strange et al., 2005*; *Tanovic et al., 2018*; *Tobia et al., 2012*). Further validation of our neuroimaging results might be needed using other techniques (e.g. task-related MRI, lesion studies) as well as replication efforts, ideally with larger samples.

One major future direction is to investigate how the mechanisms and brain networks uncovered in this study relate to AD spectrum and prospective risk of developing dementia. For example, while not needed to make a clinical diagnosis of SCI (*Jessen et al., 2014*), AD-related biological indicators (e.g. CSF biomarkers and amyloid and tau PET imaging) might provide valuable information on how processing uncertainty and information gathering relates to AD pathology in preclinical population with SCI. This could be further supported by evidence from longitudinal follow-up of individuals with SCI to establish risk factors and outcomes. Another line of research might benefit from adopting a transdiagnostic approach to examine whether and how cognitive mechanisms of information seeking and related affective dysfunction are shared (or distinguished) in different stages of AD and other forms of neurodegeneration. Similarly, examining patients who suffer from anxiety and depression without expression of cognitive complaints would help further delineate the association between affective burden and uncertainty-related behaviours.

In conclusion, our results provide evidence that hyperreactivity to uncertainty might be a key manifestation of SCI and is related to heightened functional connectivity between the insula and the hippocampus. These manifestations might be underlie affective burden in the condition.

## Materials and methods

### Participants

Twenty-seven individuals with SCI (age: $\mu = 59.81$, $SD = 7.70$, 14 females) along with 27 healthy age- and gender-matched controls (age: $\mu = 62.04 \pm SD = 6.28$) were recruited for the study. Sample size was determined based on our previous study testing and validating the behavioural paradigm in healthy young controls (*Petitet et al., 2021*) as well as comparable studies investigating information gathering in patient groups (*Clark et al., 2006*; *Hauser et al., 2017*). SCI participants were clinically assessed by trained neurologists (co-authors MH and SM) in the cognitive disorders clinic at John Radcliff Hospital, Oxford. In addition to clinical assessment, the diagnosis of SCI was supported by normal performance on standardised objective cognitive assessment using Addenbrooke's Cognitive Examination (ACE-III) with cutoff >87/100 (*Bruno and Schurmann Vignaga, 2019*; *Elamin et al., 2016*; *Hsieh et al., 2013*) as well as normal clinical MRI scan. This definition is consistent with the criteria proposed in previous key reports (*Jessen et al., 2014*; *Jessen et al., 2020*), suggesting that SCI diagnosis relies on two key components (i) subjective report of cognitive decline and (ii) normal performance on standardised objective cognitive tests.

All participants gave written consent to take part in the study and were offered monetary compensation for their participation. The study was approved by the University of Oxford ethics committee (RAS ID: 248379, Ethics Approval Reference: 18/SC/0448). *Table 1* shows demographics of the study groups. All participants completed the behavioural tasks and questionnaires. Neuroimaging data were obtained from 23 SCI participants and 25 healthy controls who were MRI compatible and gave consent to be scanned for research purposes.

### Clinical measures

All participants underwent a cognitive assessment using Addenbrooke's Cognitive Examination III (ACE-III; *Hsieh et al., 2013*). They also completed self-report questionnaires of depression and anxiety (Beck Depression Inventory II, BDI-II; *Beck et al., 1996*, and Hospital Anxiety Depression Scale, HADS; *Zigmond and Snaith, 1983*).

### Procedure

A 17-inch touchscreen PC was used to present the task, which was coded using MATLAB (The MathWorks inc, version 2018b) and Psychtoolbox version 3 (*Brainard, 1997*; *Kleiner et al., 2007*). The distance between participants and the screen was (50 cm) allowing them to reach it comfortably using their dominant hand. Task environment was adjusted according to handedness and participants were instructed to use their index finger for all their responses. An experimenter was present at all times in the testing room to answer any questions they might have.

### Experimental paradigm

In this study, we used a shorter version of *Circle Quest*, an active information gathering task that has been previously validated and extensively tested in young healthy people (explained in detail in *Petitet et al., 2021*). In this paradigm, participants were required to maximise their reward by trying to localise a hidden circle as precisely as possible. They could infer the location of the circle using clues that they could obtain by touching the screen at different locations on a designated search field (grey circle in *Figure 1*). Participants could acquire as many samples as they wanted without limitations to when and how these samples were obtained on each trial. There were two types of clues: purple dots if the location was situated inside the hidden circle, and white dots if the location was outside the circle. The sizes of hidden circle and dots were fixed on all trials (circle radius: 130 Px, 5.80% of the search space, dot radius: 4 Px). Two circles of the same size as the hidden circle were always displayed on either side of the screen in order to limit memory requirements of the task. Within these two circles was displayed the credits that participants could potentially win if they managed to localise the hidden

circle with no errors. After the information-gathering phase, participants could localise the hidden circle using a blue disc that had the same size as the hidden circle.

These aspects of the task were explained to participants using an interactive tutorial with the help of the experimenter. Following this, they performed a training task to further expose them to the task environment and its scoring rules. During training, participants were presented with different configuration of dots (four purple dots and four white dots) from which they could infer the location of the hidden circle with different levels of uncertainty (e.g. when purple dots are spaced out this indicated a lower level of uncertainty than when they were clumped close together). In this training task, participants were instructed to only move the blue disc to where they thought the hidden circle was located. Uncertainty in the task was experimentally quantified as *expected error* (*EE*) which is equal to the error an optimal agent would obtain if they placed the blue disc at the best possible location. A penalty was introduced representing how far localisation was from the true location of the hidden circle. This penalty was subtracted from credits assigned to each trial that participants could potentially win if their localisation was perfect (i.e. placing the blue disc exactly on top of the hidden circle). The penalty incurred for each error pixel was fixed on all trials and was equal to 1.2 credit/pixel, thus localisation error penalty was equal to the distance between the blue disc and hidden circle centre multiplied by 1.2. Once participants completed training, they were required to complete a task comprehension questionnaire to become eligible to continue with the behavioural task. All participants recruited for this study had no issues with this questionnaire.

## Active sampling task

In this version of the task, participants incurred costs for acquiring information: with each additional sample obtained, they lost credits from an initial credit reserve they started each trial with (i.e. from the potential reward they could win if they managed to perfectly find the location of the hidden circle using the blue disc). There were two levels of sampling cost ($\eta_s$; low: –1 credit/sample and high: –5 credits/sample) and two levels of initial credit ($R_o$; low: 95 credits, high: 130 credit/sample) giving rise to four experimental blocks (15 trials each) that were counterbalanced between participants. Each trial lasted 18 s, during which participants could sample the search field freely at any location of the search field. After 18 s, the blue disc appeared automatically and participants had 6 s to move it on top of where they thought the hidden circle was located. They then received feedback indicating the number of credits they won on the trial. This score was calculated as follows:

$$Score = R_0 - s.\eta_s - e.\eta_e \tag{1}$$

where $R_0$ is initial credit reserve, $s$ is number of samples acquired, $e$ is localisation error (the distance in pixels between the centre of the hidden circle and the centre of the blue disc), and $\eta_e$ is spatial error cost which was fixed and equal to 1.2 per pixel.

## Passive choice task

In this version of the task, participants' agency was limited. They were required to make passive decisions (accepting/rejecting offers) based on predetermined levels of uncertainty and reward for these offers. At the beginning of each trial, participants saw a configuration of dots (four purple dots and four white dots) mapping onto different experimentally defined levels of uncertainty (five levels of *EE*: 16.3–24.4, 27.1–38.9, 57.5–58.9, 73.33–74.18, 91.9–93.3 pixels). They were required to indicate how confident they are about the location of the hidden circle using a rating scale on the side of the scene. Subjective uncertainty score was calculated by z-scoring sign-flipping these confidence ratings. Following this, the reward on offer appeared in the two circles on the side of the screen (four reward levels *R*: 40, 65, 90, 115 credits). Participants were required to indicate whether they would like to place the blue disc given the reward and uncertainty of the offers. They did this by pressing 'Yes' or 'No' appearing on the screen. There were 20 different offer combinations and each offer was presented five times, thus participants completed 100 trials overall. Participants were told that after indicating their preferences, 10 of their accepted offers will be randomly selected for them to play, and that these ten offers would decide their score in the game.

## Quantifying uncertainty

Uncertainty in *Circle Quest* paradigm was quantified as the expected error (*EE*), that is the average error an ideal participant would obtain by placing the localisation disc (blue disc) at the best possible location given the dots on the screen (i.e. given the information displayed). Calculating this metric first required to compute the probability of any pixel on the screen to be the centre of the hidden circle. This was done by sequentially applying Baye's rule as described below.

The probability that a location on the screen $\lambda$ is the centre of the hidden circle given the observation $o$ ($o^+$ if the dot is purple, $o^-$ if the dot is white) at location $\sigma$ was calculated as:

$$p_s(\lambda|o,\sigma) = \frac{p_s(\lambda) \times p_s(o,\sigma|\lambda)}{p_s(o,\sigma)} \tag{2}$$

where $p_s(\lambda)$ is the prior probability, $p_s(o,\sigma|\lambda)$ is the likelihood, and $p_s(o,\sigma)$ is the probability of making the observation $o$ at the sampling location $\sigma$. Note that because Baye's rule was applied sequentially, $s$ refers here to the stage of the search, that is, the number of samples currently displayed on the screen. The posterior probability $p_s(\lambda|o,\sigma)$ becomes the prior probability $p_{s+1}(\lambda)$ for the next sample. The likelihood function was defined as:

$$\begin{cases} p_s(o^+,\sigma|\lambda) = 1 \text{ if } |\lambda - \sigma| \leq r \\ p_s(o^+,\sigma|\lambda) = 0 \text{ if } |\lambda - \sigma| > r \\ p_s(o^-,\sigma|\lambda) = 1 - p_s(o^+,\sigma|\lambda) \end{cases} \tag{3}$$

where $r$ is the radius of the hidden circle (which is constant). Finally, the probability of the observation $o$ at the sampling location $\sigma$ is the sum over all possible hidden circle centres $\lambda$ of the probability of the observation given $\lambda$, weighted by the probability of $\lambda$ to be the hidden circle centre:

$$p_s(o,\sigma) = \sum_\lambda p_s(o,\sigma|\lambda) \times p_s(\lambda) \tag{4}$$

The posterior distribution over all hidden circle locations was converted into an expected error map as follows:

$$EE_s(\lambda) = \sum_i p_s(\lambda_i) \times |\lambda - \lambda_i| \tag{5}$$

Simply put, $EE_s(\lambda)$ is the average distance between $\lambda$ and all locations on the screen, weighted by the probability of these locations to be the centre of the hidden circle. At any stage of the search, there exists an ideal circle placement location $\lambda_s^\star$ where $EE_s(\lambda_s^\star)$ is minimal. Throughout the paper, and consistent with previous work (**Petitet et al., 2021**), uncertainty was quantified as the expected error at the ideal placement location given the information on the screen (i.e. $EE_s(\lambda_s^\star)$). In the interest of simplicity, the rest of this paper uses the abbreviation *EE* to refer to this metric.

## Quantifying sampling efficiency

On average, *EE* decreased exponentially over successive samples (**Figure 2—figure supplement 1**). Because participants were free to sample anywhere on the screen, *EE* could decrease more or less sharply depending on the quality of participants' choices of sampling locations (**Petitet et al., 2021**). Sampling efficiency captures how well participants reduced *EE* from one sample to the next. It was estimated as the information extraction rate $\alpha$ by fitting the following model to individual datasets:

$$\begin{aligned} \hat{E}E_{(n,1)} &= \frac{\sum_{i=1}^{60} EE_{(i,1)}}{60} \\ \hat{E}E_{(n,s)} &= (\hat{E}E_{(n,1)} - \hat{E}E_\infty) \times (1 - \alpha_n)^{s-1} + \hat{E}E_\infty \\ 0 &< \alpha < 1 \quad \& \quad \hat{E}E_\infty > 0 \end{aligned} \tag{6}$$

where $n$ is the trial number, $s$ is the sample number and $\hat{E}E_\infty$ is the asymptotic *EE* which reflects the limitations to uncertainty reduction imposed by the task. This simple two-parameters model was fitted using a least mean squared error procedure, implemented in MATLAB (The MathWorks inc, version 2019a).

## Quantifying deviation from optimality

Behaving rationally in the active task means stopping sampling when the expected value, *EV*, is maximal. The latter evolves dynamically with every new sample acquired, and is calculated by simply replacing the error term in the score equation (*Equation 1*) by *EE*, as follows:

$$EV(s) = R_0 - s \times \eta_s - EE(s) \times \eta_e \tag{7}$$

By using the estimated expected error, $\hat{EE}_{(n,s)}$, from *Equation 6*, we could compute the expected value at every stage of the search given the cost-benefit structure ($R_0$, $\eta_s$, $\eta_e$) of the trial and the participant's average sampling efficiency ($\alpha$). That is:

$$\hat{EV}(s) = R_0 - s \times \eta_s - \hat{EE}(s) \times \eta_e \tag{8}$$

The probability of a rational agent to stop sampling at each stage of the search could be estimated by applying a *softmax* function over all possible $\hat{EV}(s)$ values, as follows:

$$p(s) = \frac{\exp(\hat{EV}(s))}{\sum_{i=0}^{\infty} \exp(\hat{EV}(i))} \tag{9}$$

*Figure 2—figure supplement 2* provides a visualisation of such predictions. Note that they account for inter-individual differences in sampling efficiency. For every participant and experimental conditions, the optimal number of samples $s^\star$ was defined as the number of samples at which $\hat{EV}(s)$ was maximal (i.e. highest probability of the rational agent to stop). By definition, acquiring less or more samples than $s^\star$ results in lower average scores than the optimal expected value. We quantified deviations from optimality as the difference between the number of samples participants actually acquired and $s^\star$.

## Statistical analyses

Statistical analysis was performed either in MATLAB R2019a or R version 4.0.2. Data from active and passive tasks were analysed mainly using generalised mixed effects models with full randomness using *fitglme* function in MATLAB. These were either logistic or linear models depending on the response variable. Full description and statistical details of these models can be found in *Supplementary Files*. Post-hoc follow-up analyses were conducted using appropriate statistical tests of difference (student t-test or Wilcoxon rank-sum test) depending on whether parametric assumptions were met. Principal component analysis was applied to HADS anxiety and BDI-II questionnaire scores using *pca* function in MATLAB. The first component of this PCA analysis was used as a measure of affective burden. All Correlations were performed using Spearman's non-parametric testing within the two groups separately, controlling for possible confounds such as age, gender, and cognitive score when required.

## Magnetic Resonance data acquisition

Structural and functional magnetic resonance scans were obtained using 3T Siemens Verio scanner at John Radcliffe Hospital, Oxford. Structural images were T1-weighted with 1 mm isotropic voxel resolution (MPRAGE, field of view: 208 × 256 × 256 matrix, TR/TE = 200/1.94ms, lip angle = 8°, iPAT = 2, prescan-normalise). Resting-state functional MRI (rfMRI) measures spontaneous changes in blood oxygenation (BOLD signal) due to intrinsic brain activity. rfMRI images had voxel size = 2.4 × 2.4 ×2.4 mm (GE-EPI with multi-band acceleration factor = 8, field of view: 88 × 88 × 64 matrix, TR/TE = 735/39ms. flip angle = 52°, fat saturation, no iPAT). MR images were obtained from 23 SCI participants and 25 healthy controls who were MRI compatible or consented to have MR scans as part of the study.

## Magnetic resonance data processing and analysis

Resting-state connectivity analysis was conducted in MATLAB 2019b using CONN toolbox v20.b (*Whitfield-Gabrieli and Nieto-Castanon, 2012*) running SPM12. Default processing pipeline was used. This included functional realignment and unwarp, slice-timing and motion correction, segmentation, and normalisation to MNI space. To increase signal-to-noise ratio, spatial smoothing was applied using spatial convolution with Gaussian kernel of 8 mm full width half maximum. Denoising was done using linear regression of potential confounds and temporal band-pass filtering (0.008–0.09 Hz). This linear regression controlled for noise components from white matter and cerebrospinal

fluid areas (16 parameters each). Head motion was controlled for using 12 noise parameters, which included three translation and three rotation parameters in addition to their first-order derivatives. Confounding effects arising from identified outliers and from linear BOLD signal trends were also controlled for.

Following this, whole-brain ROI-to-ROI was run using 40 atlas-defined ROIs. These ROIs included the default nodes in CONN used to run network functional connectivity analysis as they represent key nodes in brain networks including salience, default mode, sensorimotor, visual, dorsal attention, frontoparietal, language, and cerebellar networks. In addition, limbic brain regions including the hippocampus, para-hippocampus and amygdala were added as these regions have been consistently implicated in processing uncertainty (*Harrison et al., 2006*; *Morriss et al., 2019*; *Rigoli et al., 2019*). Overall, this analysis examined 780 connections, controlling for age and gender differences. Significance testing was done using Threshold Free Cluster Enhancement (TFCE) with corrected connection significance threshold equal to 0.05.

## Acknowledgements

We thank Maria Raquel Maio, Sofia Toniolo, Michele Veldsman, and all members of the Cognitive Neurology Research group at the University of Oxford for their assistance in some elements of data collection. This work was supported by the Wellcome Trust (206330/Z/17/Z). BA was funded by a Rhodes Scholarship. PP and MH were funded by the Wellcome Trust (206330/Z/17/Z). SGM was funded by an MRC Clinician Scientist Fellowship (MR/P00878/X). The funders had no role in study design, data collection and analysis, decision to publish or preparation of the manuscript.

## Additional information

### Funding

| Funder | Grant reference number | Author |
| --- | --- | --- |
| Wellcome Trust | 206330/Z/17/Z | Masud Husain |
| Rhodes Scholarships | | Bahaaeddin Attaallah |
| Medical Research Council | MR/P00878/X | Sanjay G Manohar |

The funders had no role in study design, data collection and interpretation, or the decision to submit the work for publication. For the purpose of Open Access, the authors have applied a CC BY public copyright license to any Author Accepted Manuscript version arising from this submission.

### Author contributions

Bahaaeddin Attaallah, Conceptualization, Data curation, Formal analysis, Investigation, Methodology, Project administration, Visualization, Writing - original draft; Pierre Petitet, Conceptualization, Formal analysis, Methodology, Writing - review and editing; Elista Slavkova, Vicky Turner, Youssuf Saleh, Data curation; Sanjay G Manohar, Supervision, Writing - review and editing; Masud Husain, Conceptualization, Funding acquisition, Resources, Supervision, Writing - review and editing

### Author ORCIDs

Bahaaeddin Attaallah  http://orcid.org/0000-0002-7842-7974
Pierre Petitet  http://orcid.org/0000-0003-1422-5326

### Ethics

Human subjects: All participants gave written consent to take part in the study. The study was approved by the University of Oxford ethics committee (RAS ID: 248379, Ethics Approval Reference: 18/SC/0448).

### Decision letter and Author response

Decision letter https://doi.org/10.7554/eLife.75834.sa1
Author response https://doi.org/10.7554/eLife.75834.sa2

# Additional files

## Supplementary files

• Transparent reporting form

• Supplementary file 1. Active Search. Generalised mixed effects models investigating the effect of the group (SCI vs. controls) on performance. Models were specified as follows: Response variable ~ 1 + group* $\eta$ + group*$R_0$ + $\eta_s$*$R_0$ + group: $\eta_s$ :$R_0$ + (1 + $\eta_s$*$R_0$ |participant). SCI: subjective cognitive impairment. $R_0$ : initial credit reserve. $\eta_s$ : sampling cost. EE: uncertainty before committing to decisions.

• Supplementary file 2. Active Search. Number of samples obtained per condition. $R_0$ : initial credit reserve. $\eta_s$ : sampling cost.

• Supplementary file 3. Active Search. Deviation from optimal ($s-s*$) for each condition. $R_0$ : initial credit reserve. $\eta_s$ : sampling cost. : number of samples obtained. s*: optimal number of samples.

• Supplementary file 4. Active Search. Generalised mixed effects models investigating the effects ofthe group (SCI vs. controls) and affective burden on uncertainty estimation. Models were specified as follows: Subjective Uncertainty ~ 1 + *group* *EE*+ (1 + *EE* |participant) + (1 |trial). Subjective Uncertainty ~ 1 + Burden* *EE*+ age + ACE-III +(1 + *EE* |participant) + (1 |trial). *EE*: Experimentally defined expected error. ACE-III: Addenbrook's Cognitive Examination score.

• Supplementary file 5. Passive choices. Generalised mixed effects models investigating effects of group (SCI vs. controls) and affective burden effect on passive decision making under uncertainty. Models were specified as follows: choice ~ 1 + group*R + group*EE + R*EE + group:R:EE + (1 + R*EE |participant). Choice ~ 1 + Burden*R + Burden*EE + R*EE + Burden:R:EE + Age + ACE-III (1 + R*EE|participant). R: reward on offer. *EE*: expected localisation error. SCI: subjective cognitive impairment group. ACE-III: Addenbrook's Cognitive Examination score.

• Supplementary file 6. Active Search. Generalised mixed effects models investigating the effect of the group (SCI vs. controls) on sampling speed (ISI) and efficiency (α). Models were specified as follows: Inter-sampling Interval (ISI) ~ 1 + group* $\eta_s$ + group*$R_0$ + $\eta_s$*$R_0$ + group: $\eta_s$ :$R_0$ + (1 + $\eta_s$*$R_0$ |participant); Information extraction rate (α) ~ 1 + group* $\eta_s$ + group*$R_0$ + $\eta_s$*$R_0$ + group: $\eta_s$ :$R_0$ + (1 + $\eta_s$*$R_0$ |participant).

• Supplementary file 7. Active Search – Inter-sampling interval per condition. $R_0$ : initial credit reserve. $\eta_s$ : sampling cost.

• Supplementary file 8. Active Search. Speed efficiency trade-off. Models were specified as follows: α ~ 1 + *ISI* + (1 +*ISI* |participant) + (1 + *ISI* |condition) + (1 |trial).

• Supplementary file 9. ROI to ROI resting functional connectivity. PaHC: para-hippocampus. Hipp: hippocampus. IC: insular cortex. unc: uncorrected. FWE: family-wise error. FDR: false discovery rate.

• Supplementary file 10. ROI to ROI resting functional connectivity with potential outliers (three SCI individuals and one control with values above or below *Q*3 + 1.5*IQR*) excluded. PaHC: parahippocampus. Hipp: hippocampus. IC: insular cortex. unc: uncorrected. FWE: family-wise error. FDR: false discovery rate.

• Supplementary file 11. Motion parameters and other quality control measures. (**a**) Six motion parameters were used during realignment procedure for rfMRI processing. These correspond to six timeseries containing three transnational and three rotational parameters over time for each subject. None of these parameters was significantly different between SCI and controls groups (all $p_{corr}$ > 0.61). (**b**) Five quality control estimates were used during preprocessing of neuroimaging data (*Whitfield-Gabrieli and Nieto-Castanon, 2012*). These included number of valid scans after scrubbing procedure, mean and maximum motion (extracted from the six parameters above), mean and maximum global signal change. None of these parameters was significantly different between the two groups (all $p_{corr}$ > 0.16 ). Based on mean motion and mean global signal changes, four potential outliers (three SCI participants and one control with values above or below $Q3 + 1.5IQR$) were identified. A second version of neuroimaging analysis was performed with these participants excluded (*Supplementary file 10*). There were no changes to the results or conclusions made in the paper. These findings suggest that rfMRI differences between SCI participants and controls are unlikely due to motion artifacts. Mean and max motion was calculated based on *Power et al., 2012*. Error bars show ± 95% CI.

• Supplementary file 12. Quality control parameters do not correlate with task measures and affective burden. Across study participants, no correlation was found between mean motion (or global signal change) and hyperreactivity to uncertainty (*ISI* or Deviation from optimal) or affective

burden (all $p_{corr} = 1$). Specifically, no correlation between *ISI* (the measure that correlates with insular-hippocampal connectivity) and these quality control measures (mean motion and mean GS change) across SCI participants ($p = 0.13$ & $p = 0.49$, respectively). These findings suggest that correlation between *ISI* and insular-hippocampal connectivity is unlikely due to motion artifacts. Correlations were controlled for age and gender. Error bars show ± 95% CI.

## Data availability

Anonymised data and code for replicating the main results in the manuscript have been deposited on the Open Science Framework platform: https://osf.io/7ysqu/.

The following dataset was generated:

| Author(s) | Year | Dataset title | Dataset URL | Database and Identifier |
|---|---|---|---|---|
| Attaallah B, Petitet P, Slavkova S, Turner V, Saleh Y, Manohar S, Husain M | 2021 | Raw and processed behavioural data from Circle Quest task in SCI and controls | https://osf.io/7ysqu/ | Open Science Framework, 7ysqu |

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

## Appendix 1

### Computational model of sampling speed, efficiency and extent

To further characterise active sampling behaviour, we fitted a previously validated computational model that also accounts for a putative hidden cognitive cost in addition to the economic costs imposed by the task (*Petitet et al., 2021*). Participants are assumed to incur a cognitive cost at each step of their search that is a function of how efficiently and quickly they sample. The model calculates the expected utility of a sample ($EU_s$) given these costs and returns five parameter estimates per participant. The first two parameters represent the weights participants assign to sample costs ($w_s$) and benefits ($w_e$). The other three parameters describe the cognitive cost function in terms of a penalty for sampling speed ($w_{speed}$) and efficiency ($w_\alpha$), as well as an intercept ($w_0$). The model was specified as follows:

$$EU_s(ISI, \alpha, t_{max}) = EU_{s-1} + p(s|ISI, t_{max}).[w_e.\eta_e \times (1 - \alpha).(EE_{s-1} - \hat{EE}_\infty) - w_s.\eta_s^{1+s} - \eta_c(ISI, \alpha)]$$

Previous EU + Probability of acquiring the sample given the current time (10)

$$. \left[ \text{Expected information benefit} - \text{Sampling cost} - \text{Cognitive effort cost} \right]$$

where $\eta_c(ISI, \alpha)$ is a cognitive cost function. $\eta_e$ is the placement error penalty and $t_{max}$ is the allowed search time per trial. These two variables were fixed ($\eta = 1.2$ credits/pixel, $t_{max} = 18$ seconds) in the version of the task used in the study. $\hat{EE}_\infty$ is the information sampling asymptotic limit, which was estimated for each individual beforehand to take into consideration inter-individual variations in asymptotic information sampling performance.

Based on previous work (*Petitet et al., 2021*), the following quadratic cognitive cost function was used:

$$\eta_c(ISI, \alpha) = w_0 + w_{speed} \times \frac{1}{ISI^2} + w_\alpha \times \alpha^2 \qquad (11)$$

The likelihood function was obtained by applying a *softmax* function over the 3-dimensional space of $EU$ ($EU$ depends on $ISI, \alpha, s$) for a given task condition, as follows:

$$p_s(stop|ISI, \alpha, t_{max}) = \frac{\exp(EU_s(ISI, \alpha, t_{max}))}{\sum_i \sum_a \sum_t^{t_{max}} \exp(EU_s(i, a, t))} \qquad (12)$$

For each individual, model fitting involved findings the parameters that achieved the lowest negative log-likelihood of observing the multivariate distribution of number of samples acquired ($s$), inter-sampling interval (ISI) and sampling efficiency ($\alpha$).

Parameter optimisation was performed in MATLAB (The MathWorks inc, version 2019a) using Bayesian Adaptive Direct Search (BADS ; *Acerbi and Ma, 2017*). Further information about this modelling framework is provided in *Petitet et al., 2021*.

### Analysis of the model's parameter estimates

The analysis of computational model parameters was in agreement with the analysis of raw behaviour reported in the main text. SCI participants and controls did not significantly differ in the decision weight they assigned to sampling cost and benefit during active search ($p = 0.80$ & $p = 0.48$, respectively; *Figure 1*). This is an alternative way to parameterise the fact that extensive sampling in SCI was not driven by task-related economic considerations. By contrast, the cognitive cost function of SCI participants differed from those of controls. SCI participants assigned a lower subjective penalty to sampling speed and efficiency (Group difference; $w_{speed}$, $w_\alpha$ : $z = 2.12$, $p = 0.03$; *Figure 1*). This explains why they performed the task along a different speed-efficiency trade-off from controls. The intercept term $w_0$ – which captures a constant subjective cost associated with sampling actions (e.g. motor cost) – did not differ between SCI participants and controls ($p = 0.80$).

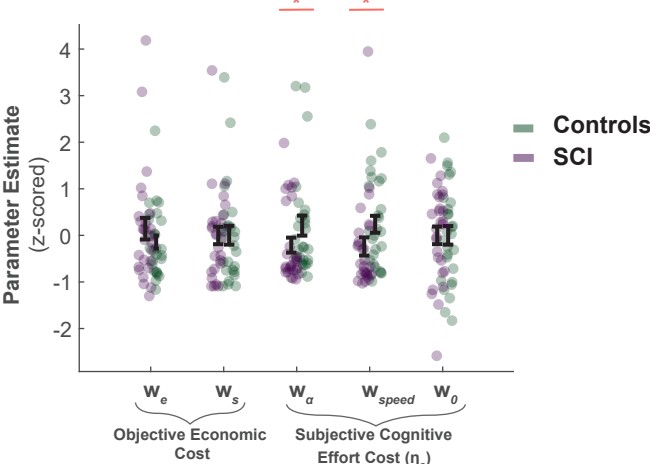

**Appendix 1—figure 1.** SCI participants assign lower costs to speed and efficiency. There was no significant difference in the weights SCI participants assigned to sampling benefit ($w_e$) or cost ($w_s$), suggesting the differences in active sampling between the two groups were unlikely due to economic constraints of the task. On the other hand, individuals with SCI had lower weights assigned to efficiency ($w_\alpha$) and speed ($w_{speed}$), indicating a lower cognitive cost to engage in faster and efficient sampling. $w_0$ captures a subjective fixed cost of sampling that is not explicitly specified in the task (e.g. cost of the motor action). This was not significantly different between the two groups.$*: p < 0.05$

