## [Editor Report]

This study tests the hypothesis that subjective cognitive impairment (SCI) is linked to hyperreactivity to uncertainty. Using an information-gathering task, the authors demonstrate that individuals with SCI sample faster and more than controls under uncertainty. The reported findings provide important new clues about the psychological and neural manifestations of SCI.

---

## [Decision Letter]

**Decision letter after peer review:**

Thank you for submitting your article "Hypersensitivity to uncertainty is key feature of subjective cognitive impairment" for consideration by *eLife*. Your article has been reviewed by 2 peer reviewers, and the evaluation has been overseen by Valentin Wyart as Reviewing Editor and Christian Büchel as the Senior Editor. The following individual involved in the review of your submission has agreed to reveal their identity: Sander Nieuwenhuis (Reviewer #1).

The reviewers have discussed their reviews with one another, and the Reviewing Editor has drafted this to help you prepare a revised submission. As you will see, both reviewers have found that your study provides important new clues about the psychological and neural manifestations of subjective cognitive impairment (SCI), and we agree with this evaluation. However, the two reviewers have also identified a number of essential revisions, detailed below, that they agreed should be addressed in a point-by-point response letter, submitted together with the revised manuscript. Individual reviews are provided at the bottom of this message for your information, they do not require point-by-point responses. We hope that you will be able to address these different points in a revised version of your manuscript.

Essential revisions:

1) The test of overestimation or over-use of uncertainty in the passive task is limited by its reliance on self-report measures. Indeed, it is established that self-report confidence does not necessarily map neatly onto a single internal uncertainty variable that is used to inform decision-making (see, e.g., Navajas, 2017, Nature Human Behaviour). This means that it is not possible using the used task design to rule out alterations in uncertainty estimation in SCI. If the authors do not have an empirical way to deal with this limitation of the task design, they should at least mention this limitation explicitly in a paragraph of the Discussion section.

2) The use of the term 'hypersensitivity' throughout the manuscript to characterize the main finding is problematic. First, if the SCI group was hypersensitive to uncertainty, then why did they not differ from controls in how they estimated and valued uncertainty in the 'passive' condition? In other places in the manuscript, the authors summarize the same key findings as the SCI group being 'hyperreactive' to uncertainty. Given the importance of agency and active sampling, the term 'hyperreactivity' seems far more appropriate than 'hypersensitivity' to characterize the main finding. Second, the term 'hypersensitivity' lacks a level of explanation – what exactly is 'hypersensitivity'? And why should it specifically result in faster and more exhaustive information? The authors should either clarify more explicitly upfront in the manuscript what is meant by 'hypersensitivity', or even better adjust their terminology by switching to 'hyperreactivity', including in the title of the paper.

3) Important information about psychometrics is missing in the current version of the manuscript. First, while the use of PCA to generate a measure of general affective burden is sensible, it ignores potential differential effects of anxiety and depression. The authors should therefore investigate and report the effects of each separately to clarify whether behavioral tendencies are indeed related to a common affective component rather than to specific symptoms. Second, were subjects assessed for clinical anxiety and depression? If so, this could be mentioned. Third, was the level of SCI quantified by any means? And if so, did it relate to anxiety/depression or sampling behavior? And last, was there any correlation between affective burden and confidence or acceptance probability in the passive task? This should also be reported in the revised manuscript.

4) Alternative accounts of the main finding are missing. Even if the proposed interpretation of the main finding is appealing, there are alternative explanations that are hard to exclude conclusively, in particular related to subjective costs (regardless of objective values in the task). For example, it may be that the subjective cost of not finding the circle (e.g., the personal cost of failing at the task) is exaggerated in SCI, which could explain the more extensive sampling, or potentially a lower subjective cost of sampling. Such alternative accounts should at least be discussed explicitly in a dedicated paragraph in the Discussion section.

5) Cross-subject correlations between a number of variables that differ between the two groups are highly problematic when pooling across the two groups. If you take two variables that both significantly differ between groups and then correlate the two variables across all study participants, you are almost bound to find a significant correlation, even when within each group there is no significant correlation. This problem is illustrated by the classic statistical textbook example of a spurious correlation between two random point clouds that are separated from each other along both axes. These correlations (gray numbers in Figures 5C and 7) should therefore not be reported because they probably reflect little more than the significant main effects of the group on the two variables that were already reported in earlier parts of the Results section. The fact that the correlations between measures of hyperreactivity and affective burden (purple numbers in Figure 5C) are also significant when only the SCI participants are included is reassuring. The same is true for the correlation between hyperreactivity and insular-hippocampal connectivity (purple numbers in Figure 7). The problem is with the correlation between affective burden and insular-hippocampal connectivity (Figure 7): this correlation is significant when all participants are included but is clearly not present when only the SCI participants (or only the controls) are considered. This means that it cannot be concluded that these measures are related to each other. The same line of argument also invalidates the mediation analysis that includes all three variables (and corresponding conclusions). If the authors want to report a mediation analysis in the revised manuscript, they should base it only on the SCI participants. Reviewer #1 has noted that the authors acknowledge that "Caution should be exercised when interpreting these [correlation] results as these models were performed with data from both groups included. This was done to take advantage of the larger sample size for this type of analysis, while also trying to explore inter-individual differences regardless of group assignment." However, based on the statistical issue raised above, this is not sufficient. These between-subject correlation analyses including all participants should be removed from the manuscript, and the authors should adjust the Results and Discussion sections accordingly. As stressed by Reviewer #1, there are plenty of remaining valid results that are interesting and novel once the problematic analyses have been removed from the paper.

6) Were there any group differences in motion parameters from the imaging data? Or any correlations with symptoms or behavior in the task? These analyses would be helpful to confirm that there are no issues with motion that could explain the connectivity results.

7) How does one determine the optimal number of samples? This quantification should be unpacked much more explicitly in the revised manuscript.

8) Have there been any previous fMRI studies of functional connectivity in SCI? If so, then their results should be summarized somewhere in the Discussion section, and compared to the current results.

9) The authors should either make explicit what the voxel-wise seed-based functional connectivity analysis adds to the whole-brain analysis or remove it from the revised manuscript. Otherwise, readers may see this additional analysis as a form of 'double-dipping'.

---

## [Author Response]

Essential revisions:1) The test of overestimation or over-use of uncertainty in the passive task is limited by its reliance on self-report measures. Indeed, it is established that self-report confidence does not necessarily map neatly onto a single internal uncertainty variable that is used to inform decision-making (see, e.g., Navajas, 2017, Nature Human Behaviour). This means that it is not possible using the used task design to rule out alterations in uncertainty estimation in SCI. If the authors do not have an empirical way to deal with this limitation of the task design, they should at least mention this limitation explicitly in a paragraph of the Discussion section.

We agree that estimation of uncertainty in the passive task is limited to self-reported confidence. This is now addressed in the Discussion section as follows:

“It should also be noted that the measure of subjective uncertainty based on self-reports in the passive may map imperfectly onto actual internal uncertainty (Navajas et al., 2017; Peters, 2022). Self-reports of confidence might be contaminated by other factors such as perceptual uncertainty (e.g., meaning of perceived stimuli on the task), memory (e.g., remembering task instructions or previous estimations) and decisions (e.g., confidence in estimation). The contribution of these metacognitive dimensions could be explored in future studies, especially given that SCI – by definition – is characterised by inconsistency between self-report estimates and objective measures (Jessen et al., 2020). Similarly, SCI might involve different cognitive domains (e.g., memory, attention, language, etc.) with different levels of severity (Diaz-Galvan et al., 2021; Jessen et al., 2020; Miebach et al., 2019; Si et al., 2020; Smart et al., 2014; Si et al., 2020; Diaz-Galvan et al., 2021; Miebach et al., 2019). How such differences relate to uncertainty processing and affective burden will require a fine-grained approach aimed specifically at characterising SCI cognitive complaints in-depth to disentangle the dimensions involved. ”

However, we also note that there is an important distinction between over-estimation and over-use of uncertainty, and the test of the latter is not totally reliant on self-report of uncertainty. Use (valuation) of uncertainty was measured by analysing the decisions requiring a trade-off between uncertainty and reward (passive task). Here, the term ‘sensitivity’ might be appropriate (see below – reviewers’ comment 2) as we are interested in how SCI participants change their acceptance based on changes in expected error (EE) and reward on offer. Participants with a steeper slope of acceptance as a function of changes in EE (or reward) are considered to have higher sensitivity to changes in uncertainty (or reward) compared to participants with flatter slopes. In the generalised mixed effects model used to analyse the data, this is equal to an interaction term between group and EE/reward (group X EE/reward), which was not significant – visually evident by almost identical slopes of offer acceptance as function of reward and EE. Author response image 1 displays another visualisation of these sensitivity measures extracted from the model. Note that there was no significant difference between SCI individuals and controls, indicating similar incentivisation by both attributes when making decisions (see Figure 3c. and Table in Supplementary file 5). Thus, there was no evidence of over-use of uncertainty in SCI participants based on these objective measures extracted from participants’ choices.

**Author response image 1. sa2fig1:** Intact passive decision making in SCI. The weights that participants assigned to reward and uncertainty in the passive task were extracted from the generalised mixed effects model and plotted for visualisation. These represent the degree to which people change their acceptance as a function of change in the offer attribute (uncertainty and reward). There was no significant difference between SCI individuals and controls, indicating similar incentivisation by both attributes when making decisions..

2) The use of the term 'hypersensitivity' throughout the manuscript to characterize the main finding is problematic. First, if the SCI group was hypersensitive to uncertainty, then why did they not differ from controls in how they estimated and valued uncertainty in the 'passive' condition? In other places in the manuscript, the authors summarize the same key findings as the SCI group being 'hyperreactive' to uncertainty. Given the importance of agency and active sampling, the term 'hyperreactivity' seems far more appropriate than 'hypersensitivity' to characterize the main finding. Second, the term 'hypersensitivity' lacks a level of explanation – what exactly is 'hypersensitivity'? And why should it specifically result in faster and more exhaustive information? The authors should either clarify more explicitly upfront in the manuscript what is meant by 'hypersensitivity', or even better adjust their terminology by switching to 'hyperreactivity', including in the title of the paper.

We have changed the term ‘hypersensitivity’ to ‘hyperreactivity’ throughout the manuscript, including the title.

3) Important information about psychometrics is missing in the current version of the manuscript. First, while the use of PCA to generate a measure of general affective burden is sensible, it ignores potential differential effects of anxiety and depression. The authors should therefore investigate and report the effects of each separately to clarify whether behavioral tendencies are indeed related to a common affective component rather than to specific symptoms.

We have now added the results for anxiety and depression separately to the revised manuscript.

“To assess the individual contribution of anxiety and depression to these findings in SCI, the same analysis was performed separately for each affective component. This showed that BDI-II score (depression) correlated with faster sampling (R^2^ = 0.17, p = 0.039) but not over-sampling (p = 0.15) in SCI. The HADS score (anxiety) on the other hand correlated with over-sampling (R^2^ = 0.17, p = 0.038) but not speed (p = 0.16). However, these effects became insignificant (p = 0.12 for both) when the two affective components partialled out each other (partial correlations), suggesting that while there might be distinctive effects of depression and anxiety, these effects are better explained by the general affective burden imposed by these two dimensions given their association.”

Second, were subjects assessed for clinical anxiety and depression? If so, this could be mentioned.

During the clinical interview, all participants were assessed for anxiety and depression but not by a psychiatrist with expertise in making such diagnoses, so we report simply objective questionnaire scores.

Third, was the level of SCI quantified by any means? And if so, did it relate to anxiety/depression or sampling behavior?

No. level of SCI was not quantified. This is an important question that should be investigated in future research. We highlighted this in the revised discussion:

“Similarly, SCI might involve different cognitive domains (e.g., memory, attention, language, etc.) with different levels of severity (Diaz-Galvan et al., 2021; Jessen et al., 2020; Miebach et al., 2019; Si et al., 2020; Smart et al., 2014; Si et al., 2020; Diaz-Galvan et al., 2021; Miebach et al., 2019). How such differences relate to uncertainty processing and affective burden will require a fine-grained approach aimed specifically at characterising SCI cognitive complaints in-depth to disentangle the dimensions involved.”

“Additionally, because of the hippocampus’ well-established mnemonic function, insular-hippocampal processing of uncertainty might also affect one’s awareness of memory performance, giving rise to subjective cognitive complaints characterising SCI. Testing such a hypothesis will require future research with a detailed examination of the nature of subjective complaints, their severity, and association with uncertainty and expression of concerns.”

And last, was there any correlation between affective burden and confidence or acceptance probability in the passive task? This should also be reported in the revised manuscript.

We investigated this effect using two additional LMMs for confidence and decisions. There was no significant effect of affective burden in these two models. These results have now been added to the revised manuscripts and tables with summary statistics are added to supplementary tables (Tables in Supplementary files 4 and 5).

“Finally, the correlation between affective burden and passive task performance was investigated using LMMs. The results showed that affective burden had no significant effect (main or interaction) on subjective uncertainty or acceptance probability (Main effect of affective burden on subjective uncertainty: β = −0.003, t(5394) = −0.27, p = 0.79; Main effect of affective burden on offer acceptance: β = −0.151, t(5390) = −1.96, p = 0.05; See Supplementary files 4 and 5 for full statistical details including interactions)”

4) Alternative accounts of the main finding are missing. Even if the proposed interpretation of the main finding is appealing, there are alternative explanations that are hard to exclude conclusively, in particular related to subjective costs (regardless of objective values in the task). For example, it may be that the subjective cost of not finding the circle (e.g., the personal cost of failing at the task) is exaggerated in SCI, which could explain the more extensive sampling, or potentially a lower subjective cost of sampling. Such alternative accounts should at least be discussed explicitly in a dedicated paragraph in the Discussion section.

Alternative accounts of the main finding are now developed more extensively in the discussion (see text below). In one of these paragraphs, we highlight the computational model used to capture some of the agent-related costs in active sampling. Results from this model are included in Supplementary information. We chose not to add these results to the main text for the sake of simplicity as they represent alternative formalisation of the results already reported (relying on harder concepts to grasp for the reader). Interested readers can consult Appendix 1 for full report.

“One might argue that rather than suggesting that SCI individuals are hyperreactive to uncertainty in the active task, our findings might alternatively be explained by a lower reactivity to reward – as participants lost more credits to obtain the extra information. However, some observations in the study suggest that this might not be the case. First, the influence of economic constraints (R_0_ , η_s_ ) on sampling behaviour and speed did not significantly differ between individuals with SCI and controls. If SCI participants were less reactive to reward, then one would expect economic constraints to affect sampling behaviour in SCI participants to a lesser degree than age- and gender-matched controls. Instead, the group effect (i.e., acquiring more samples in individuals with SCI) was equally manifested in all experimental conditions, regardless of the current cost-benefit structure. Second, acquiring more samples indeed allowed SCI participants to achieve lower uncertainty levels prior to decision, suggesting that these additional samples carried instrumental utility and were not merely reflective of wasteful sampling behaviour driven by insensitivity to reward.

Another account of SCI individuals’ extensive sampling behaviour might involve subjective costs that are not directly specified in economic terms in the task. Namely, SCI individuals and controls might differ in the subjective cost they assign to sampling speed and efficiency. As demonstrated previously, these additional agent-related factors contribute to the utility of the samples being acquired and thus influence the extensiveness of sampling (Petitet et al., 2021). Given that SCI participants sampled faster but as efficiently as controls, these subjective costs appear to be lower in this population (see Appendix 1 for a formal demonstration using a computational model). As a result, samples carry higher utility overall, which promotes more extended data gathering. Potentially, there are several other subjective costs that people might be considering when gathering information. For example, SCI participants might have exaggerated cost linked to bad performance or inaccurate placements of the circle. We aimed to limit this specific cost by not displaying the hidden circle as feedback and showing the credits won or lost instead (Figure 1). In addition, since the task depends on touching the screen to obtain clues, the cost of these motor actions might have influenced sampling behaviour (Carland et al., 2019; Cisek and Kalaska, 2010; Morel et al., 2017; Pierrieau et al., 2021; Rangel and Hare, 2010; Scott, 2012). It is possible, for example, that SCI participants might be regarding reaching and touching the screen as less effortful than controls or assigning lower costs for motor precision. While we tried to limit the influence of the motor aspects on performance by strictly controlling the settings (e.g., fixed distance from the screen, using only index finger of dominant hand), the contribution of motor elements to the observed behaviour cannot be entirely determined or excluded and might pose a limitation. Such inter-individual differences could be in part captured by the intercept term in the computational model used to fit active information sampling data (w_0_), which was not significantly different between the two groups (see Appendix 1).

In a similar vein, while the results from the passive task suggest that SCI individuals have intact estimation of uncertainty, this might not necessarily apply to a dynamic situation such as the active task. The distinct contexts of the two tasks (passive viewing vs. active gathering) may impose different computations involved in uncertainty estimation. For example, SCI individuals and controls might specifically differ in how they judge self-generated configurations of uncertainty (active task) vs. externally predefined ones (passive task) (Acosta, 1982; Gaschler et al., 2014; Kemper et al., 2012; Acosta, 1982;; Kemper and Gaschler, 2017). People might also differ in the reference points (temporal or spatial) they compare updated environments with as more information is added (Koop and Johnson, 2012; O’Donoghue and Sprenger, 2018; Sher and McKenzie, 2006; Tversky and Kahneman, 1986). It is also possible that these computations might be cognitively effortful, influencing how frequently or/and efficiently people re-estimate and update uncertainty during their search (Bhui et al., 2021; Lieder and Griffiths, 2019). All these factors might have contributed with different degrees to the pattern of the results observed in the study and might be a limitation to active information sampling investigations in general.”

5) Cross-subject correlations between a number of variables that differ between the two groups are highly problematic when pooling across the two groups. If you take two variables that both significantly differ between groups and then correlate the two variables across all study participants, you are almost bound to find a significant correlation, even when within each group there is no significant correlation. This problem is illustrated by the classic statistical textbook example of a spurious correlation between two random point clouds that are separated from each other along both axes. These correlations (gray numbers in Figures 5C and 7) should therefore not be reported because they probably reflect little more than the significant main effects of the group on the two variables that were already reported in earlier parts of the Results section. The fact that the correlations between measures of hyperreactivity and affective burden (purple numbers in Figure 5C) are also significant when only the SCI participants are included is reassuring. The same is true for the correlation between hyperreactivity and insular-hippocampal connectivity (purple numbers in Figure 7). The problem is with the correlation between affective burden and insular-hippocampal connectivity (Figure 7): this correlation is significant when all participants are included but is clearly not present when only the SCI participants (or only the controls) are considered. This means that it cannot be concluded that these measures are related to each other. The same line of argument also invalidates the mediation analysis that includes all three variables (and corresponding conclusions). If the authors want to report a mediation analysis in the revised manuscript, they should base it only on the SCI participants. Reviewer #1 has noted that the authors acknowledge that "Caution should be exercised when interpreting these [correlation] results as these models were performed with data from both groups included. This was done to take advantage of the larger sample size for this type of analysis, while also trying to explore inter-individual differences regardless of group assignment." However, based on the statistical issue raised above, this is not sufficient. These between-subject correlation analyses including all participants should be removed from the manuscript, and the authors should adjust the Results and Discussion sections accordingly. As stressed by Reviewer #1, there are plenty of remaining valid results that are interesting and novel once the problematic analyses have been removed from the paper.

We removed all cross-subject correlations reported in the manuscript and made the necessary changes to all sections of the revised manuscript.

6) Were there any group differences in motion parameters from the imaging data? Or any correlations with symptoms or behavior in the task? These analyses would be helpful to confirm that there are no issues with motion that could explain the connectivity results.

Six motion parameters were used during realignment procedure for rfMRI processing. These correspond to six timeseries containing three transnational and three rotational parameters over time for each subject. None of these parameters was significantly different between SCI and controls groups (all p_corr_ > 0.61). Five quality control estimates were used during reprocessing of neuroimaging data. These included number of valid scans after scrubbing procedure, mean and maximum motion (extracted from the six motion parameters), mean and max global signal change (Whitfield-Gabrieli and Nieto-Castanon, 2012). None of these parameters was significantly different between the two groups (all p_corr_ > 0.16). Based on mean motion and global signal changes, four potential outliers (three SCI participants and one control) were identified. A second version of the neuroimaging analysis was performed with these participants excluded. There were no changes to the results or conclusions made in the paper. These findings suggest that rfMRI differences between SCI participants and controls are unlikely due to motion artifacts. These analyses are now added to supplementary information in the revised manuscript (Supplementary files 10 and 11)

We also investigated correlations between mean motion or mean global signal change and ISI, over-sampling and affective burden. No significant correlation was found (all p_corr_ = 1). Specifically, no correlation between ISI and these quality control measures (mean motion and mean GS change) across SCI participants (p = 0.13 and p = 0.49, respectively). These control analyses have been added to supplementary information in the revised manuscript (Supplementary file 12).

7) How does one determine the optimal number of samples? This quantification should be unpacked much more explicitly in the revised manuscript.

We included more details about this in the methods section. As recommended by reviewer #1 we moved the equations for the quantification of uncertainty, efficiency, and optimal solution to the main method section in the revised manuscript.

The paragraph below in the revised manuscript explains quantification of optimal solution and deviation from optimality. We also summarised these predictions in Figure 2—figure supplement 2 in the revised manuscript.

“Quantifying deviation from optimality.

Behaving rationally in the active task means stopping sampling when the expected value, EV, is maximal. […] We quantified deviations from optimality as the difference between the number of samples participants actually acquired and s^⋆^.

8) Have there been any previous fMRI studies of functional connectivity in SCI? If so, then their results should be summarized somewhere in the Discussion section, and compared to the current results.

We have now included this in the Discussion section.

“These neuroimaging findings are consistent with previous fMRI studies (task-related and resting) in SCI showing altered connectivity between temporal brain regions (e.g., hippocampus and para-hippocampus) and several other areas and networks including default mode and salience networks (Cai et al., 2020; Dillen et al., 2017; Hafkemeijer et al., 2013;Hu et al., 2017; Rodda et al., 2009; Verfaillie et al., 2018; Viviano and Damoiseaux, 2020). […] Further validation of our neuroimaging results might be needed using other techniques (e.g., task-related MRI, lesion studies) as well as replication efforts, ideally with larger samples.”

9) The authors should either make explicit what the voxel-wise seed-based functional connectivity analysis adds to the whole-brain analysis or remove it from the revised manuscript. Otherwise, readers may see this additional analysis as a form of 'double-dipping'.

This analysis was performed for visualisation purposes only. To avoid double-dipping, we removed it from the revised manuscript.